# EDGE: Explaining Deep Reinforcement Learning Policies

**Wenbo Guo**
The Pennsylvania State University
wzg13@ist.psu.edu

**Xian Wu**[*]
The Pennsylvania State University
xkw5132@psu.edu

**Usmann Khan**[*]
Georgia Institute of Technology
ukhan35@gatech.edu

**Xinyu Xing**
Northwestern University
The Pennsylvania State University
xinyu.xing@northwestern.edu

## Abstract

With the rapid development of deep reinforcement learning (DRL) techniques, there is an increasing need to understand and interpret DRL policies. While recent research has developed explanation methods to interpret how an agent determines its moves, they cannot capture the importance of actions/states to a game's final result. In this work, we propose a novel self-explainable model that augments a Gaussian process with a customized kernel function and an interpretable predictor. Together with the proposed model, we also develop a parameter learning procedure that leverages inducing points and variational inference to improve learning efficiency. Using our proposed model, we can predict an agent's final rewards from its game episodes and extract time step importance within episodes as strategy-level explanations for that agent. Through experiments on Atari and MuJoCo games, we verify the explanation fidelity of our method and demonstrate how to employ interpretation to understand agent behavior, discover policy vulnerabilities, remediate policy errors, and even defend against adversarial attacks.

## 1 Introduction

Deep reinforcement learning has shown great success in automatic policy learning for various sequential decision-making problems, such as training AI agents to defeat professional players in sophisticated games [74, 65, 24, 37] and controlling robots to accomplish complicated tasks [33, 38]. However, existing DRL agents make decisions in an opaque fashion, taking actions without accompanying explanations. This lack of transparency creates key barriers to establishing trust in an agent's policy and scrutinizing policy weakness. This issue significantly limits the applicability of DRL techniques in critical application fields (*e.g.,* finance [47] and self-driving cars [11]).

To tackle this limitation, prior research (*e.g.,* [9, 13, 73]) proposes to derive an explanation for a target agent's action at a specific time step. Technically, this explanation can be obtained by pinpointing the features within the agent's observation of a particular state that contribute most to its corresponding action at that state. Despite demonstrating great potential to help users understand a target agent's

---

[*]Equal Contribution.

35th Conference on Neural Information Processing Systems (NeurIPS 2021).

individual actions, they lack the capability to extract insights into the overall policy of that agent. In other words, existing methods cannot shed light on the general sensitivity of an agent's final reward from a game in regards to the actions/states in that game episode. Consequently, these methods fall short in troubleshooting an agent's policy's weaknesses when it fails its task.

We propose a novel explanation method to derive strategy-level interpretations of a DRL agent. As we discuss later in Section 3, we define such interpretations as the identification of critical time steps contributing to a target agent's final reward from each game episode. At a high level, our method identifies the important time steps by approximating the target agent's decision-making process with a self-explainable model and extracting the explanations from this model. Specifically, given a well-trained DRL agent, our method first collects a set of episodes and the corresponding final rewards of this agent. Then, it fits a self-explainable model to predict final rewards from game episodes. To model the unique correlations in DRL episodes and enable high-fidelity explanations, rather than simply applying off-the-shelf self-explanation techniques, we develop a novel self-explainable model that integrates a series of new designs. First, we augment a Gaussian Process (GP) with a customized deep additive kernel to capture not only correlations between time steps but, more importantly, the joint effect across episodes. Second, we combine this deep GP model with our newly designed explainable prediction model to predict the final reward and extract the time step importance. Third, we develop an efficient inference and learning framework for our model by leveraging inducing points and variational inference. We refer to our method as "Strategy-level **E**xplanation of **D**rl a**GE**nts" (for short EDGE). [2]

With extensive experiments on three representative RL games, we demonstrate that EDGE outperforms alternative interpretation methods in terms of explanation fidelity. Additionally, we demonstrate how DRL policy users and developers can benefit from EDGE. Specifically, we first show that EDGE could help understand the agent's behavior and establish trust in its policy. Second, we demonstrate that guided by the insights revealed from our explanations, an attacker could launch efficient adversarial attacks to cause a target agent to fail. Third, we demonstrate how, with EDGE's capability, a model developer could explain why a target agent makes mistakes. This allows the developer to explore a remediation policy following the explanations and using it to enhance the agent's original policy. Finally, we illustrate that EDGE could help develop a defense mechanism against a newly emerging adversarial attack on DRL agents. To the best of our knowledge, this is the first work that interprets a DRL agent's policy by identifying the most critical time steps to the agent's final reward in each episode. This is also the first work that demonstrates how to use an explanation to understand agent behavior, discover policy vulnerabilities, patch policy errors, and robustify victim policies.

## 2   Related Work

Past research on DRL explanation primarily focuses on associating an agent's action with its observation at a particular time step (*i.e.,* pinpointing the features most critical to the agent's action at a specific time). Technically, these methods can be summarized in the following categories.

- **Post-training explanation** is a method that utilizes and extends post-training interpretation approaches (*e.g.,* [56, 28, 36, 35]) to derive explanation from a DRL agent's policy/value network and thus treat it as the interpretation for that DRL agent (*e.g.,* [9, 44, 32, 68, 20, 72]).

- **Model approximation** is an approach that employs a self-interpretable model to mimic the target agent's policy networks and then derives explanation from the self-interpretable model for the target DRL agent (*e.g.,* [13, 22, 55, 14, 59, 58, 87, 85]).

- **Self-interpretable modeling** is an approach different from the model approximation techniques above. Instead of mimicking the target agent's policy network, self-interpretable modeling builds a self-explainable model to replace the policy network. Since the new model is interpretable, one can easily derive an explanation for the target agent (*e.g.,* [92, 62, 82, 42]).

- **Reward decomposition** is a method that re-engineers a DRL agent's reward function to make the reward gained at each time step more meaningful and explainable. With the more meaningful reward in hand, at each time step, one could use the instant reward gain to interpret the agent's action (*e.g.,* [73, 46, 54]).

---

[2]The source code of EDGE can be found in `https://github.com/Henrygwb/edge`.

From the objective perspective, our work is fundamentally different from the above DRL explanation research. Rather than pinpointing the features – in an observation – critical for an agent's action, our work identifies the critical time steps contributing to the agent's final reward. Using our explanation, one can better understand the agent's policy, unveil policy weakness, and patch policy errors (as shown in Section 5). In Supplement S7, we further conduct a user study to demonstrate the superiority of our method against the above explanation approaches in pinpointing good policies and performing policy forensics. Note that there are two other methods that also understand a DRL policy through the agent's previous memories [49, 23]. These works are fundamentally different from ours in two perspectives. First, both methods have a different explanation goals from our work. Specifically, Koul *et al.* [49] focuses on identifying whether the action at each time step depends more on the current observation or the previous states. The method proposed in [23] pinpoints the important steps w.r.t. the subsequent transitions in the FSM extracted from the target agent rather than the final result of an episode. Second, both methods can be applied only to white-box RNN policies, whereas our method is applicable to DRL policies with arbitrary network structures.

## 3 Key Technique

### 3.1 Problem Setup

Consider a DRL game with an agent trained with Q-learning [86, 60] or policy gradient [48, 69, 70]. Our work aims to explain this agent's policy by identifying the important steps contributing most to a game episode's final reward/result. To ensure practicability, we allow access only to the environment states, agent's actions, and rewards. We assume the availability of neither the value/Q function nor the policy network. Formally, given $N$ episodes $\mathcal{T} = \{\mathbf{X}^{(i)}, y_i\}_{i=1:N}$ of the target agent, $\mathbf{X}^{(i)} = \{\mathbf{s}_t^{(i)}, \mathbf{a}_t^{(i)}\}_{t=1:T}$ is the $i-$th episode with the length $T$, where $\mathbf{s}_t^{(i)} \in \mathbb{R}^{d_s}$ and $\mathbf{a}_t^{(i)} \in \mathbb{R}^{d_a}$ are the state and action at the time step $t$. $y_i$ is the final reward of this episode.[3] Our goal is to highlight the top-K important time steps within each episode $\mathbf{X}^{(i)}$.

**Possible Solutions and Limitations.** The most straightforward approach of identifying important time steps is to use the output of the value/Q network as indicators and pinpoint the time steps with the top K highest value/Q function's outputs as the top K critical steps. However, since we do not assume the availability of these networks, this method is not applicable to our problem. A more realistic method is to fit a seq2one model (*i.e.,* RNN) that takes as input the state-action pairs in an episode and predicts the final reward of that episode. With this prediction model, one could utilize a post-training explanation method to derive the time step importance. However, existing post-training explanation techniques usually require approximating the target DNN with more transparent models, which inevitably introduces errors. Additionally, many post-training methods are vulnerable to adversarial attacks [61, 30, 94] or generate model-independent explanations [1, 64, 2, 78]. As we will show later in Section 4 and Supplement S3&S5, these limitations jeopardize the post-training explanations' fidelity. A more promising direction is to fit a self-explainable model to predict the final reward. Existing research has proposed a variety of self-explanation methods. Most of them do not apply to our problem because they either cannot derive feature importance as explanations [5, 52, 50, 18], cannot be applied to sequential data [21, 27, 12], or require explanation ground truth [16, 66]. In this work, we consider two self-explainable models that are designed to fit and explain sequential data – an RNN augmented with attention [10, 39, 34] and rationale net [51]. Technically, both models have the form of $g(\theta(\mathbf{x}) \odot \mathbf{x})$, where $\theta(\cdot)$ is a weight RNN or an attention layer and $g(\cdot)$ is the prediction RNN. The output of $\theta(\cdot)$ can be used to identify the important steps in the input sequence. Despite extracting meaningful explanations, recent research [45, 89, 17] reveals that the explanations given by $\theta(\cdot)$ cannot faithfully reflect the associations (*i.e.,* feature importance) learned by the subsequent prediction model $g(\cdot)$, leading to an even lower fidelity than the post-training explanations in some applications. Additionally, these models are not designed to explain an RL agent and cannot fully capture the dependencies within the episodes of that agent. Specifically, the episodes collected from the same agent tend to exhibit two types of dependencies: dependency between the time steps within an episode and the dependency across different episodes. Although they consider the dependency

---

[3]For the games with delayed rewards, such as MuJoCo [84] and Atari Pong [8], where a non-zero reward $r_T$ is assigned only to the last step of a game, we use $r_T$ as $y_i$. For the games with instant rewards (*e.g.,* OpenAI CartPole [15]), we compute an episode's total reward as $y_i$, i.e., $y_i = \sum_t r_t$.

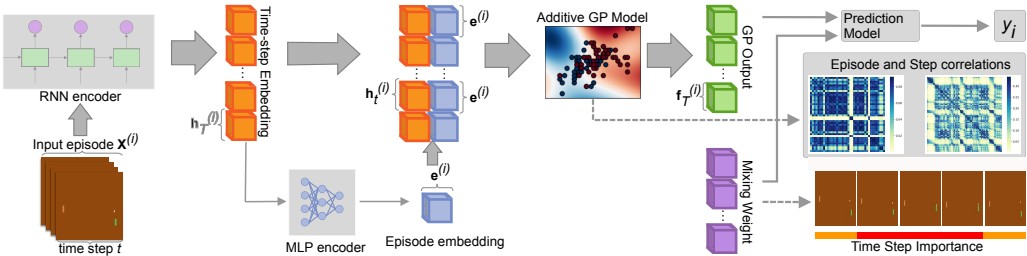

Figure 1: Overview of EDGE with a constant prediction mixing weight.

within each input sequence, these methods cannot capture the correlations between different inputs. As is shown in Section 4 and Supplement S3&S5, this also jeopardizes their explanation fidelity.

## 3.2 Explanation Model Design of EDGE

In this work, we design a novel self-explainable model by adopting a very different design than existing methods. First, to better capture the associations (*i.e.,* feature importance) learned by the prediction model, we add the explainable module to the final layer rather than the input layer of the prediction model. Formally, our model can be written as $g(f(\mathbf{x}))$, where $f(\cdot)$ is a feature extractor and $g(\cdot)$ is an explainable prediction model. Second, we design a deep Gaussian Process as the feature extractor to capture the correlations between time steps and those across different episodes, which are often exhibited in a set of episodes collected from the same DRL agent. In addition to capturing different levels of correlations, another advantage of deep GPs over typical DNNs is that GPs model the joint distribution of the output signals, enabling access to the output signals' uncertainty. Finally, we design an interpretable Bayesian prediction model to infer the distribution of final rewards and deliver time step importance. Below, we first give an overview of our proposed model. Then, we describe how to adapt the traditional GP model to our problem, followed by the design of the final prediction model.

**Overview.** As shown in Fig 1, given an episode of the target agent $\mathbf{X}^{(i)}$, EDGE first inputs it into a RNN encoder, which outputs the embedding of each time step in this episode $\{\mathbf{h}_t^{(i)}\}_{t=1:T}$. EDGE also passes the last step's embedding through a shallow MLP to obtain an episode embedding $\mathbf{e}^{(i)}$. Then, EDGE adopts our proposed additive GP framework to process $\{\mathbf{h}_t^{(i)}\}_{t=1:T}$ and $\mathbf{e}^{(i)}$ and obtains a latent representation of the whole episode $\mathbf{f}_{1:T}^{(i)}$. As introduced later, this representation is able to capture the correlations between time steps and those across episodes. Finally, EDGE inputs $\mathbf{f}_{1:T}^{(i)}$ into our prediction model $\mathbf{f}_{1:T}^{(i)}$ and get the predicted final reward of the input episode. As detailed later, our prediction model is designed based on a linear regression, whose regression coefficient can be used to identify important time steps within in the input episode.

**Additive GP with Deep Recurrent Kernels.** Gaussian Process defines a distribution over an infinite collection of random variables, such that any finite subset of variables follows a multivariate Gaussian distribution [63]. In Statistical modeling, GP defines the prior of a non-parametric function $f : \mathcal{X} \to \mathbb{R}$. Formally, if $f$ has a GP prior, i.e., $f \sim \mathcal{GP}(0, k_\gamma)$, where $k_\gamma(\cdot, \cdot)$ is a positive semi-definite kernel function parameterized by $\gamma$, any finite collections of $\mathbf{f} \in \mathbb{R}^N$ follows a multivariate Gaussian distribution $(\mathbf{f}|\mathbf{X}) \sim \mathcal{N}(\mathbf{0}, K_{XX})$. Here, $K_{XX} \in \mathbb{R}^{N \times N}$ is the covariance matrix, with $(K_{XX})_{ij} = k_\gamma(\mathbf{x}_i, \mathbf{x}_j)$. In our model, we adopt the widely applied square exponential (SE) kernel function: $k_\gamma(\mathbf{x}_i, \mathbf{x}_j) = \exp\left(-\frac{1}{2}(\mathbf{x}_i - \mathbf{x}_j)^T \Theta_k (\mathbf{x}_i - \mathbf{x}_j)\right)$, with $\gamma = \Theta_k$. Traditional GP with SE kernel [63] assumes the input space is Euclidean, which is usually invalid for real-world data with high-dimensional inputs [3]. To tackle this challenge, recent research [91, 53] proposes to conduct dimensional reduction via a DNN and then apply a GP to the DNN's latent space. They show that the resultant deep kernel models achieve similar performance to DNNs on complicated datasets.

In our model we capture the sequential dependency within an episode by using an RNN as the deep net inside the kernel function. Specifically, given an episode $\mathbf{X}^{(i)}$, we first concatenate the state and action (*i.e.,* $\mathbf{x}_t^{(i)} = [\mathbf{s}_t^{(i)}, \mathbf{a}_t^{(i)}]$), input them into an RNN $h_\phi$, and obtain the latent representation of this episode: $\{\mathbf{h}_t^{(i)}\}_{t=1:T}$, where $\mathbf{h}_t^{(i)} \in \mathbb{R}^q$ is the state-action embedding at the time $t$. We also

compute an episode embedding by passing the last step's hidden representation through a shallow MLP $e_{\phi_1} : \mathbf{h}_T^{(i)} \to \mathbf{e}^{(i)} \in \mathbb{R}^q$. After obtaining $\{\mathbf{h}_t^{(i)}\}_{t=1:T}$ and $\mathbf{e}^{(i)}$, we then adopt the additive GP framework to capture the correlations between time steps and those across episodes. Formally, an additive GP is the weighted sum of $J$ independent GPs, i.e., $f = \sum_J \alpha_j f_j$. Here, $f_j \sim \mathcal{GP}(0, k_j)$ is the $j$-th GP component, in which the covariance function $k_j$ is typically applied to a subset of input features. By assigning every component a GP prior, one can ensure that the mixed-signal $f$ also follows a GP prior [25]. Following this framework, we construct our deep GP model as the sum of two components $f_t$ and $f_e$. Specifically, $f_t \sim \mathcal{GP}(0, k_{\gamma_t})$ models the correlations between time-steps, where the covariance between the $t$-th steps in episode $i$ and the $k$-th steps in episode $j$ can be computed by $k_{\gamma_t}(\mathbf{h}_t^{(i)}, \mathbf{h}_k^{(j)})$. Going beyond modeling the correlations between individual steps, $f_e \sim \mathcal{GP}(0, k_{\gamma_e})$ captures a higher level cluster structures within the collected episodes, i.e., the similarity between episodes. Formally, the episode-level covariance between any pair of time steps in episode $i$ and $j$ is given by $k_{\gamma_e}(\mathbf{e}^{(i)}, \mathbf{e}^{(j)})$. Finally, our deep additive GP model can be expressed as: $f = \alpha_t f_t + \alpha_e f_e$, where $\alpha_t$ and $\alpha_e$ are the component weights. Given a set of collected episodes represented by $\mathbf{T} \in \mathbb{R}^{N \times T \times (d_s + d_a)}$, $\mathbf{f} \in \mathbb{R}^{NT}$ is given by: $\mathbf{f} | \mathbf{X} \sim \mathcal{N}(\mathbf{0}, k = \alpha_t^2 k_{\gamma_t} + \alpha_e^2 k_{\gamma_e})$, where $\mathbf{X} \in \mathbb{R}^{NT \times (d_s + d_a)}$ is the flattened matrix of $\mathbf{T}$.

**Prediction Model.** To ensure explanability, we use a linear regression as the base of our prediction model, where the regression coefficients reflect the importance of each input entity. Specifically, we first convert the flattened response $\mathbf{f}$ back to the matrix form $\mathbf{F} \in \mathbb{R}^{N \times T}$, where the $i$-th row $\mathbf{F}^{(i)} \in \mathbb{R}^T$ is the $i$-th episode's encoding given by our GP model. Then, we define the conditional likelihood for the discrete and continuous final reward, respectively. When $y_i$ is continuous, we follow the typical GP regression model [63] and define the $y_i = \mathbf{F}^{(i)} \mathbf{w}^T + \epsilon_1$, where $\mathbf{w} \in \mathbb{R}^{1 \times T}$ is the mixing weight and $\epsilon_1 \sim \mathcal{N}(0, \sigma^2)$ is the observation noise. The conditional likelihood distribution is $y_i | \mathbf{F}^{(i)} \sim \mathcal{N}(\mathbf{F}^{(i)} \mathbf{w}^T, \sigma^2)$. For the discrete final reward with a finite number of possible values, we use the softmax prediction model to perform classification. Formally, we define $y_i | \mathbf{F}^{(i)}$ follows a categorical distribution with $p(y_i = k | \mathbf{F}^{(i)}) = \frac{\exp((\mathbf{F}^{(i)} \mathbf{W}^T)_k)}{\sum_k \exp((\mathbf{F}^{(i)} \mathbf{W}^T)_k)}$. $\mathbf{W} \in \mathbb{R}^{K \times T}$ is the mixing weight, where $K$ is the total number of classes. Finally, we combine all the components together and write our explanation model as (A illustration of our proposed model can be found in Fig. 1.): [4]

$$\mathbf{f} | \mathbf{X} \sim \mathcal{N}(\mathbf{0}, k = \alpha_t^2 k_{\gamma_t} + \alpha_e^2 k_{\gamma_e}), \quad y_i | \mathbf{F}^{(i)} \sim \begin{cases} \text{Cal}(\text{softmax}(\mathbf{F}^{(i)} \mathbf{W}^T)), & \text{If conducting classification} \\ \mathcal{N}(\mathbf{F}^{(i)} \mathbf{w}^T, \sigma^2), & \text{otherwise} \end{cases},$$
(1)

where the mixing weight is constant. This indicates the time step importance derived from the mixing weight is a global explanation. [5] According to the insight that time steps with a high correlation tend to have a joint effect (similar importance) on the game result, we could combine the global explanation with the time step correlations in $K_t(\mathbf{X}, \mathbf{X})$ to gain a fine-grained understanding of each game. Specifically, given an episode and the top important steps indicated by the mixing weight, we can identify the time steps that are highly correlated to these globally important steps and treat them together as the local explanation of that episode. Supplement S1 introduces another way of deriving episode-specific explanations by replacing the constant mixing weight with a weight obtained by a simple DNN. Note that the episode correlations in $K_e(\mathbf{X}, \mathbf{X})$ reveal the cluster structure within a set of episodes, which helps categorize the explanations of similar episodes.

### 3.3 Posterior Inference and Parameter Learning

**Sparse GP with Inducing Points.** Direct inference of our model requires computing $(K_{XX} + \sigma^2 I)^{-1}$ over $K_{XX}$, which incurs $\mathcal{O}(NT^3)$ computational complexity. This cubic complexity restricts our model to only small datasets. To improve scalability, we adopt the inducing points method [91] for inference and learning. At a high level, this method simplifies the posterior computation by reducing the effective number of samples in $\mathbf{X}$ from $NT$ to $M$, where $M$ is the number of inducing points. Specifically, we define each inducing point at the latent space as $\mathbf{z}_i \in R^{2q}$, and $\mathbf{u}_i$ as the GP output

---

[4]Note that our model is similar to existing GP-based state-space models [4, 71, 29, 81, 26] in that both use an RNN inside the kernel function. However, these models do not integrate an additive GP. More importantly, their prediction models are not designed for explanation purposes and thus cannot derive time step importance.

[5]The classification model gives $K$ global explanations, one for each class derived from each row of $\mathbf{W}$.

of $\mathbf{z}_i$. Then, the joint prior of $\mathbf{f}$ and $\mathbf{u}$ and the conditional prior $\mathbf{f}|\mathbf{u}$ are given by:

$$\mathbf{f}, \mathbf{u}|\mathbf{X}, \mathbf{Z} \sim \mathcal{N}\left(\begin{bmatrix} \mathbf{0} \\ \mathbf{0} \end{bmatrix}, \begin{bmatrix} K_{XX} & K_{XZ} \\ K_{XZ}^T & K_{ZZ} \end{bmatrix}\right), \mathbf{f}|\mathbf{u}, \mathbf{X}, \mathbf{Z} \sim \mathcal{N}(K_{XZ}K_{ZZ}^{-1}\mathbf{u}, K_{XX} - K_{XZ}K_{ZZ}^{-1}K_{XZ}^T), \quad (2)$$

where $K_{XX}$, $K_{XZ}$, $K_{ZZ}$ are the covariance matrices. They can be computed by applying our additive kernel function to the time-step and episode embedding of the training episodes and inducing points. As is shown in Eqn (2), with inducing points, we only need to compute the inverse of $K_{ZZ}$, which significantly reduces the computational cost from $\mathcal{O}(NT^3)$ to $\mathcal{O}(m^3)$.

**Variational Inference and Learning.** So far, our model has introduced the following parameters: neural encoder parameters $\Theta_n = \{\phi, \phi_1\}$, GP parameters $\Theta_k = \{\gamma_t, \gamma_e, \alpha_e, \alpha_t\}$, prediction model parameters $\Theta_p = \{\mathbf{w}/\mathbf{W}, \sigma^2\}$, and inducing points $\mathbf{Z} = \{\mathbf{z}_i\}_{i=1:M}$. To learn these parameters, we follow the idea of empirical Bayes [63] and maximize the log marginal likelihood $\log p(y|\mathbf{X}, \mathbf{Z}, \Theta_n, \Theta_k, \Theta_p)$. Maximizing this log marginal likelihood is computationally expensive and, more important, intractable for models with non-Gaussian likelihood. To provide factorized approximation to marginal likelihood and enable efficient learning, we assume a variational posterior over the inducing variable $q(\mathbf{u}) \sim \mathcal{N}(\mu, \Sigma)$ and a factorized joint posterior $q(\mathbf{f}, \mathbf{u}) = q(\mathbf{u})p(\mathbf{f}|\mathbf{u})$, where $p(\mathbf{f}|\mathbf{u})$ is the conditional prior in Eqn. (2). By Jensen's inequality, we can derive the evidence lower bound (ELBO):

$$\log p(y|\mathbf{X}, \mathbf{Z}, \Theta_n, \Theta_k, \Theta_p) \geq \mathbb{E}_{q(\mathbf{f})}[\log p(y|\mathbf{f})] - \text{KL}[q(\mathbf{u})||p(\mathbf{u})], \quad (3)$$

where the first part is the likelihood term. The second KL term penalizes the difference between the approximated posterior $q(\mathbf{u})$ and the prior $p(\mathbf{u})$. Maximizing the ELBO in Eqn. (3) will automatically maximize the marginal likelihood, which is also equivalent to minimizing the KL divergence from the variational joint posterior to the true posterior (See Supplement S1 for more details).

When conducting classification, the categorical likelihood makes the likelihood term in Eqn. (3) intractable. To tackle this challenge, we first compute the marginal variational posterior distribution of $\mathbf{f}$, denoted as $q(\mathbf{f}) = \mathcal{N}(\mu_f, \Sigma_f)$ (See Supplement S1 for detailed computations). Then, we apply the reparameterization trick [57] to $q(\mathbf{f})$. Formally, we define $\mathbf{f} = v(\epsilon_f) = \mu_f + \mathbf{L}_f\epsilon_f$, with $\epsilon_f \sim \mathcal{N}(\mathbf{0}, \mathbf{I})$ and $\mathbf{L}_f\mathbf{L}_f^T = \Sigma_f$. With this reparameterization, we can sample from the standard Gaussian distribution and approximate the likelihood term with Monte Carlo (MC) method [83]:

$$\mathbb{E}_{q(\mathbf{f})}[\log p(y|\mathbf{f})] = \mathbb{E}_{p(\epsilon_f)}[\log p(y|v(\epsilon_f))] \approx \frac{1}{B} \sum_b \sum_i \log p(y_i|(\mathbf{F}^{(i)})^{(b)}), \quad (4)$$

where $B$ is the number of MC samples. For the regression model, we directly compute the analytical form of likelihood term and use it for parameter learning (See derivation in Supplement S1). With the above approximations, our model parameters (*i.e.*, $\Theta_n$, $\Theta_k$, $\Theta_p$, $\mathbf{Z}$, and $\{\mu, \Sigma\}$) can be efficiently learned by maximizing the (approximated) ELBO using a stochastic gradient descent method. Implementation details and hyper-parameter choices can be found in Supplement S2.

## 4 Evaluation

In this section, we evaluate `EDGE` on three representative RL games (all with delayed rewards) – Pong in Atari, You-Shall-Not-Pass in MuJoCo, and Kick-And-Defend in MuJoCo. Supplement S5 further demonstrates the effectiveness of our method on two OpenAI GYM games (both with instant rewards). For each game, we used a well-trained agent as our target agent (See Supplement S2 for more details about these agents).

**Baseline Selection.** Recall our goal is to take as input the episode of a target agent and identify the steps critical for the agent's final reward. As is discussed in Section 3.1, to do this, there are two categories of alternative approaches – ❶ fitting an episode through a non-interpretable model and then deriving explanation from that model and ❷ fitting an episode through a self-explainable model and then obtaining interpretation directly from its interpretation component. In this section, we select some representative alternative methods as our baseline and compare them with our proposed method. Below, we briefly describe these baseline approaches and discuss the rationale behind our choice.

With respect to the first type of alternative approaches, we first utilize the RNN structure proposed in [43] to fit the reward prediction model. Then, we apply various gradient-based saliency methods on the RNN model and thus derive interpretation accordingly. We implement three widely used saliency

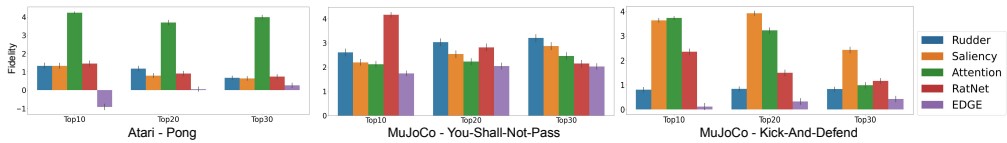

Figure 2: Mean and standard error of the fidelity scores obtained by each explanation method. The x-axis represents the different choices of $K$. "RatNet" stands for Rationale Net. For our method, we use the global explanations derived from the mixing weight in this evaluation.

methods – Vanilla gradient [76], integrated gradient [80], and SmoothGrad [77] – as well as their variants (ExpGrad [79], VarGrad [40], and integrated gradient with uniform baseline [79]).[6] When comparing RNN+saliency method with our proposed approach, we choose the RNN's interpretation from the saliency method with the best explanation fidelity. For the fidelity comparison between each saliency method, the readers could refer to Supplement S3. In addition to the RNN+saliency method, another method falling into the first kind of alternative approaches is Rudder [7]. Technically speaking, this method also learns an RNN model to predict an agent's final reward. Differently, it derives explanation from decomposed final reward.

Regarding the second kind of alternative approaches, we choose Attention RNN and Rationale Net. Attention RNN [10] is typically treated as a self-interpretable model. From the model's attention layer, one could extract its output and use it as the important scores for the input dimensions. We use these important scores to pinpoint the critical time steps in the input episode. Similar to Attention RNN, Rationale Net is also self-interpretable. In our experiments we use Rationale Net's original model structure [51] rather than the improved model structure proposed in [17]. This is because, going beyond training data, the improved model training requires additional information, which is unavailable for our problem.

**Evaluation Metric.** An intuitive method to evaluate the fidelity of the various approaches' explanations is to vary the actions at the time steps critical for the final reward and then measure the reward difference before and after the action manipulation. However, this method invalidates the physical realistic property of an episode because the change of an agent's action at a specific time step would inevitably influence its consecutive actions and the state transitions. To address this problem, we introduce a physically realistic method to manipulate episodes. Then, we introduce a new metric to quantify the fidelity of interpretation.

Given the explanation of the $i$-th episode – $\mathbf{E}_i$, we first identify the top-$K$ important time steps from $\mathbf{E}_i$. From the top-$K$ time steps, we then extract the longest sequence (*i.e.,* the longest continuous time steps), record its length – $l$, and treat its elements as the time steps most critical to $y_i$.

To evaluate and compare the fidelity of the interpretation (*i.e.,* the most critical time steps extracted through different interpretation methods), we first replay the actions recorded on that episode to the time step indicated by the longest sequence. Starting from the beginning of the longest continuous time steps to its end (*i.e.,* $t_i \cdots t_{i+l}$), we replace the corresponding actions at these time steps with random actions. [7] Following the action replacement, we pass the state at $t_{i+l+1}$ to the agent's policy. Starting from $t_{i+l+1}$, we then use the agent's policy to complete the game, gather the final reward, and compute the final reward difference before/after replaying denoted as $d$. After computing $l$ and $d$, we define the fidelity score of $\mathbf{E}_i$ as $\log(p_l) - \log(p_d)$. Here, $p_l = l/T$ is the length of the longest sequence normalized by the total length of the episode - $T$. $p_d = |d|/d_{\max}$ is the absolute reward difference normalized by the maximum absolute reward difference of the game. When the value of the fidelity score $\log(p_l) - \log(p_d)$ is low, it indicates $\mathbf{E}_i$ is illustrated by a short length of sequence. By varying the actions pertaining to this short sequence, we can observe a great change in the agent's final reward. As such, a low score implies high fidelity of an interpretation method.

**Result.** Fig. 2 shows the comparison results of EDGE against the aforementioned alternative explanation approaches. First, we observe that existing self-explainable methods (*i.e.,* Attention and Rational

---

[6]Note that we select these saliency methods because they pass the sanity check [1, 2]. Besides, it should be noted that we do not consider the perturbation-based methods to derive interpretation from RNN because these methods are mainly designed to explain convolutional networks trained for image recognition tasks.

[7]If the policy network is an RNN, we also fit the observation at time $t_i \cdots t_{i+l}$ into the policy to ensure the RNN policy's memory is not truncated.

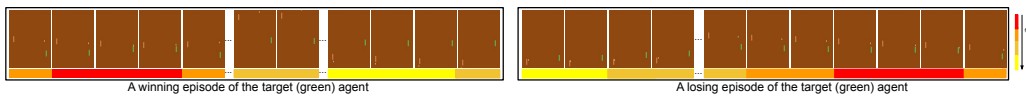

(a) Time step importance of the target agent in the Pong game.

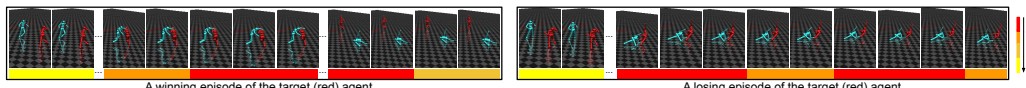

(b) Time step importance of the target regular agent in the You-Shall-Not-Pass game.

Figure 3: Illustrations of the critical time steps extracted by EDGE in a winning/losing episode.

Table 1: The target agent's performance in different use cases. "MuJoCo-Y" represents the You-Shall-Not-Pass game and "MuJoCo-K" stands for the Kick-And-Defend game. To demonstrate the statistical significance of our results, we run all the experiments three times with different random seeds and show the mean and standard error of results on each setup. Numbers before the brackets are means and those in the brackets are standard deviations. Supplementary S6 further shows a hypothesis test result.

| Applications | Games | Rudder | Saliency | Attention | RatNet | EDGE |
|---|---|---|---|---|---|---|
| Target agent win rate changes before/after attacks | Pong | -19.93 (4.43) | -30.33 (0.47) | -25.27(1.79) | -29.20 (4.24) | **-65.47 (2.90)** |
| | MuJoCo-Y | -32.53 (4.72) | -29.33 (3.68) | -33.93 (5.77) | -30.00 (1.63) | **-35.13 (2.29)** |
| | MuJoCo-K | -21.80 (3.70) | -37.87 (6.31) | -41.20 (4.70) | -7.13 (2.50) | **-43.47 (4.01)** |
| Target agent win rate changes before/after patching | Pong | +1.89 (1.25) | -1.13 (0.96) | -0.58 (1.81) | -3.66 (1.35) | **+2.75 (0.65)** |
| | MuJoCo-Y | +1.76 (0.17) | +0.92 (0.32) | +0.44 (0.06) | +1.68 (0.50) | **+2.91 (0.32)** |
| | MuJoCo-K | +0.96 (0.1) | +1.17 (0.17) | +0.57 (0.04) | **+1.21 (0.16)** | **+1.21 (0.13)** |
| Victim agent win rate changes before/after robustifying | MuJoCo-Y | +8.54 (0.75) | +12.69 (1.46) | +25.10 (1.44) | +25.42 (1.32) | **+35.30 (3.02)** |

Net) cannot consistently outperform the post-training explanation approaches (*i.e.,* saliency methods and Rudder). This observation aligns with our discussion in Section 3.1. Second, we discover that our method demonstrates the highest interpretation fidelity across all the games in all settings. As we discuss in Section 3.2, it is because our method could capture not only the inter-relationship between time steps but, more importantly, the joint effect across episodes.

In addition to the fidelity of our interpretation, we also evaluate the stability of our explanation and measure the explainability of each approach with regard to the underlying model. We present the experiment results in Supplement S3, demonstrating the superiority of our method in those dimensions. Along with this comparison, we further describe how well our method could fit given episodes, discuss the efficiency of our proposed approach, and test its sensitivity against the choice of hyper-parameters. Due to the page limit, we also detail these experiments and present experimental results in Supplement S3.

## 5   Use Cases of Interpretation

**Understanding Agent Behavior.** Fig. 3 showcases some episode snapshots of the target agent in the Pong and You-Shall-Not-Pass game together with the time-step importance extracted by our method. As we can first observe from Fig. 3(a), in the winning (left) episode, EDGE highlights the time steps when the agent hits the ball as the key steps leading to a win. This explanation implies that our target agent wins because it sends a difficult ball bouncing over the sideline and sailing to the corner where the opponent can barely reach. Oppositely, our method identifies the last few steps that the target agent misses the ball as the critical step in the losing episode. This indicates that the agent loses because it gets caught out of position. Similarly, our method also pinpoints the critical time steps matching human perceptions in the You-Shall-Not-Pass games. For example, in the left episode of Fig. 3(b), our explanations state that the runner (red agent) wins because it escapes from the blocker and crosses the finish line. Overall, Fig. 3 demonstrates that the critical steps extracted by EDGE can help humans understand how an agent wins/loses a game. In Supplement S4, we show more examples of critical time steps and the correlations we extracted from the three games. Supplement S7 further shows user study to demonstrate that our explanation could help user understand agent behaviors and thus perform policy forensics.

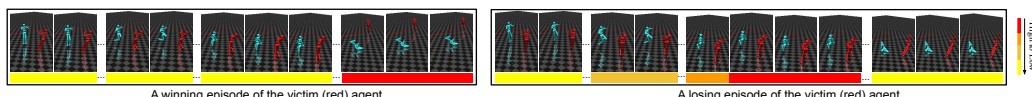

A winning episode of the victim (red) agent    A losing episode of the victim (red) agent

Figure 4: Time step importance of the victim agent in the You-Shall-Not-Pass game. By comparing this figure with Fig. 3(b), we can observe that our method could derive different explanations for different policies in the same game, indicating explanation is policy dependent.

**Launching Adversarial Attacks.** The qualitative analysis above reveals that an agent usually wins because of its correct moves at the crucial steps. With this finding, we now discuss how to launch adversarial attacks under the guidance of the interpretation. Previous research [41, 93] has proposed various attacks to fail a DRL agent by adding adversarial perturbation to its observations at each time step. We demonstrate that with the help of the explanations, an attacker could defeat an agent by varying actions at only a few critical steps rather than adding physically unrealistic perturbations. Our key idea is intuitive. If an agent's win mainly relies on its actions at a few crucial steps, then the agent could easily lose if it takes sub-optimal actions at those steps. Guided by this intuition, we propose an explanation-based attack that varies the agent's action at the critical steps identified by an explanation method. To test this attack's effectiveness, we first collect 2000 episodes where the target agent wins and explain these episodes with EDGE and the baseline approaches. Second, we conclude the top-$K$ commonly critical steps across all the episodes (Here, we set $K$=30). Finally, we run the agent in the environment and force it to take a random action at the common important steps. We test the agent for 500 rounds and record the changes in its winning rate before/after attacks in Table 1. As we can observe from the 2∼4 row of Table 1, all the explanation models can generate effective attacks that reduce the agent's winning rate. Benefiting from the high explanation fidelity, the attack obtained from our explanations demonstrates the strongest exploitability. Supplement S4 shows the results of different choices of $K$ and discusses the potential alternatives of our attack. Note that this attack is different from the fidelity test in Section 4 in that our attack generalizes the summarized time step importance to unseen episodes while the fidelity test replays the explained episodes.

**Patching Policy Errors.** We design an explanation-guided policy patch method. The key idea is to explore a remediation policy by conducting explorations at the critical time steps of losing games and use the mixture of the original policy and the remediation policy as the patched policy. Specifically, we first collected a set of losing episodes of the target agent and identified the important time steps with EDGE and the baseline approaches above. Then, we explore the remediation policy by replay those episodes with different actions at the critical steps. Here, since we do not assume an oracle knowing the correct actions to take, we perform random explorations. First, we set an exploration budget of 10, representing replaying 10 times for each losing episode. In each replaying, we take a random action at the top 5 consecutive critical steps and record the random actions and corresponding states leading to a win. Finally, we form a look-up table with these collected state-action pairs and use it as the remediation policy. When running in the environment, the target agent will act based on the table if the current state is in the table. [8] Otherwise, the agent will take the actions given by its original policy. To test the effectiveness of our method, we run 500 games and record the changes in the target agent's winning rate before and after patching. As is shown in row 5∼7 of Table 1, overall, the patched policies enhance the target agent's performance, and EDGE demonstrates the highest winning rate improvement. Table 1 also shows that in some cases, the patched policy introduces too many false positive that even reduce the winning rate. In Supplement S4, we discuss how to mitigate this problem via a probabilistic mixture of the remediation policy and the original policy. Supplement S4 also experiments the influence of the look-up table size on the patching performance and discusses other alternatives to our patching method.

**Robustifying Victim Policies.** Finally, we apply our methods to explain the episodes of a victim agent playing against an adversarial opponent in the You-Shall-Not-Pass game. The adversarial policy is obtained by the attack proposed in [31]. Fig. 4 demonstrates the identified important steps. First, the losing episode in Fig. 4 shows the blocker takes a sequence of adversarial behaviors (*i.e.,* intentionally falling on the ground). These malicious actions trick the runner into falling and thus losing the game.

---

[8] For games with a continuous state space, we compute the $l_1$ norm difference of the current state $s_t$ and the states $s_i$ in the table. If the state difference is lower than a small threshold (1e-4 in our experiment. We tested 1e-3, 1e-4, and 1e-5 and observed similar results.), we treat $s_t$ and $s_i$ as the same state. Since the games of the same agent usually start from relatively similar states and transition following the same policy, it is possible to observe similar states in different episodes.

Oppositely, the similar adversarial actions in the winning episode cannot trigger the runner to behave abnormally. The explanations reveal that the different focus of the victim causes the different victim actions. In the winning episode, the victim agent focuses less on the steps pertaining to adversarial actions, whereas those steps carry the highest weights in the losing episode. This finding implies that the victim agent may be less distracted by the adversarial actions if it does not observe them. Guided by this hypothesis, we propose to robustify the victim agent by blinding its observation on the adversary at the critical time steps in the losing episode (*i.e.*, the time steps pertaining to adversarial actions). We test the partially blind victim and record the changes in its winning rate before/after blinding. As is shown in the last row of Table 1, blinding the victim based on our explanations significantly improves its winning rate. Table 1 also demonstrates the effectiveness of the baseline approaches in robustifying victim policies. Overall, we demonstrate that the explanations of a victim policy could pinpoint the root cause of its loss and help develop the defense mechanism.

## 6   Discussion

**Scalability.** As is discussed in Section 3.3, by using inducing points and variational inference, our model parameters can be efficiently solved by stochastic gradient descents. Supplement S3&S5 show that EDGE imposes only a small training overhead over the existing methods. We can further accelerate the training of EDGE by leveraging more advanced matrix computation methods, such as approximating the covariance matrix with kernel structure interpolation [90] or replacing Cholesky decomposition with Contour Integral Quadrature when computing the $K_{ZZ}^{-1}$ [67].

**Other games.** Besides the two-party Markov games (*i.e.*, Atari Pong and MuJoCo) studied in this work, many other games also have delayed rewards – mainly multi-player Markov games (*e.g.*, some zero-sum real-time strategy games [88]) and extensive-form games (*e.g.*, Go [74] and chess [75]). Regarding the multi-player Markov games, the associations between the episodes and final rewards will also be more sophisticated, requiring a model with a high capacity to fit the prediction. As part of future work, we will investigate how to increase the capacity of our proposed model for those games, such as adding more GP components or using a more complicated DNN as the mixing weight. For the extensive-form games, only one agent can observe the game state at any given time step and thus take action. As such, these games have a different form of episodes from the Markov games. In the future, we will explore how to extend our model to fit and explain the episodes collected from extensive-form games. Supplement S5 demonstrates our method's explanation fidelity on two games with instant rewards. Future work will evaluate the effectiveness of our attack and patching methods on those games and generalize our method to more sophisticated games with instant rewards.

**Limitations and Future Works.** Our work has a few limitations. First, we mainly compare EDGE with some existing techniques that have been used to explain sequential data. It is possible that with some adaptions, other explanation methods can also be applied to sequential data. It is also possible that existing methods can be extended to capture the correlations between episodes. As part of future work, we will explore these possibilities and broader solutions to explain a DRL policy. Second, the fidelity evaluation method introduced in Section 4 could be further improved, such as identifying multiple continuous important sequences. Our future work will investigate more rigorous fidelity testing methods and metrics. Third, our current learning strategy provides the point estimate of the mixing weight (explanations). In future work, we will explore how to place a prior on the model parameters and apply Bayesian inference (*e.g.*, MCMC [6]) to output the explanation uncertainty. Finally, our work also suggests that it may be possible to train a Transformer on MDP episodes to analyze offline trajectory data [19], and then add a GP on top to perform ablation studies. As part of future works, we will explore along this direction.

## 7   Conclusion

This paper introduces EDGE to derive strategy-level explanations for a DRL policy. Technically, it treats the target DRL agent as a blackbox and approximates its decision-making process through our proposed self-explainable model. By evaluating it on three games commonly utilized for DRL evaluation, we show that EDGE produces high-fidelity explanations. More importantly, we demonstrate how DRL policy users and developers could benefit from EDGE to understand policy behavior better, pinpoint policy weaknesses, and even conduct automated patches to enhance the original DRL policy.

## Acknowledgments

We would like to thank the anonymous reviewers and meta reviewer for their helpful comments. This project was supported in part by NSF grant CNS-2045948 and CNS-2055320, by ONR grant N00014-20-1-2008, by the Amazon Research Award, and by the IBM Ph.D. Fellowship Award.

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
