# EDGE: Explaining Deep Reinforcement Learning Policies

## S1 Additional Technical Details

**Evidence Lower Bound.** In the following, we derive the evidence lower bound (ELBO) in the Eqn. (3) of Section 3.3 and explain why maximizing it is equivalent to minimizing the KL divergence from the variational joint distribution to the true posterior. Specifically, we start with the log marginal likelihood $\log p(y|\mathbf{X}, \mathbf{Z})$ and show how to derive ELBO from it

$$
\begin{aligned}
\log p(y|\mathbf{X}, \mathbf{Z}) &= \log \int\int p(y, \mathbf{f}, \mathbf{u}|\mathbf{X}, \mathbf{Z})\mathrm{d}\mathbf{u}\mathrm{d}\mathbf{f} \\
&= \log \int\int p(y|\mathbf{f}, \mathbf{X})p(\mathbf{f}, \mathbf{u}|\mathbf{X}, \mathbf{Z})\mathrm{d}\mathbf{u}\mathrm{d}\mathbf{f} \\
&= \log \int\int p(y|\mathbf{f}, \mathbf{X})p(\mathbf{f}, \mathbf{u}|\mathbf{X}, \mathbf{Z})\frac{q(\mathbf{f}, \mathbf{u})}{q(\mathbf{f}, \mathbf{u})}\mathrm{d}\mathbf{u}\mathrm{d}\mathbf{f} \\
&= \log \int\int p(y|\mathbf{f}, \mathbf{X})q(\mathbf{f}, \mathbf{u})\frac{p(\mathbf{f}, \mathbf{u}|\mathbf{X}, \mathbf{Z})}{q(\mathbf{f}, \mathbf{u})}\mathrm{d}\mathbf{u}\mathrm{d}\mathbf{f} \\
&= \log \mathbb{E}_{q(\mathbf{f}, \mathbf{u})}[p(y|\mathbf{f}, \mathbf{X})\frac{p(\mathbf{f}, \mathbf{u}|\mathbf{X}, \mathbf{Z})}{q(\mathbf{f}, \mathbf{u})}] \\
&\overset{(a)}{\geq} \mathbb{E}_{q(\mathbf{f}, \mathbf{u})}[\log\,(p(y|\mathbf{f}, \mathbf{X})\frac{p(\mathbf{f}, \mathbf{u}|\mathbf{X}, \mathbf{Z})}{q(\mathbf{f}, \mathbf{u})})] \\
&\geq \mathbb{E}_{q(\mathbf{f}, \mathbf{u})}[\log p(y|\mathbf{f}, \mathbf{X}) + \log \frac{p(\mathbf{f}, \mathbf{u}|\mathbf{X}, \mathbf{Z})}{q(\mathbf{f}, \mathbf{u})}] \\
&\geq \mathbb{E}_{q(\mathbf{f}, \mathbf{u})}[\log p(y|\mathbf{f}, \mathbf{X})] - \mathbb{E}_{q(\mathbf{f}, \mathbf{u})}[\log \frac{q(\mathbf{f}, \mathbf{u})}{p(\mathbf{f}, \mathbf{u}|\mathbf{X}, \mathbf{Z})}] \\
&\geq \mathbb{E}_{q(\mathbf{f}, \mathbf{u})}[\log p(y|\mathbf{f})] - \mathbb{E}_{q(\mathbf{f}, \mathbf{u})}[\log \frac{p(\mathbf{f}|\mathbf{u})q(\mathbf{u})}{p(\mathbf{f}|\mathbf{u})p(\mathbf{u})}] \\
&\geq \mathbb{E}_{q(\mathbf{f}, \mathbf{u})}[\log p(y|\mathbf{f})] - \int p(\mathbf{f}|\mathbf{u})\mathrm{d}\mathbf{f}\int[\log \frac{q(\mathbf{u})}{p(\mathbf{u})}]q(\mathbf{u})\mathrm{d}\mathbf{u} \\
&\geq \mathbb{E}_{q(\mathbf{f})}[\log p(y|\mathbf{f})] - \mathbb{KL}[q(\mathbf{u})||p(\mathbf{u})]\,,
\end{aligned}
\tag{1}
$$

where we omit the parameters $\{\Theta_n, \Theta_k, \Theta_p\}$ and $q(\mathbf{u})$ is the variational distribution. The $(a)$ step is according to the Jensen's inequality. As we can observe from Eqn. (1), maximizing the ELBO will automatically maximize the marginal likelihood. Below, we derive the ELBO from the KL divergence from $q(\mathbf{f}, \mathbf{u})$ to $p(\mathbf{f}, \mathbf{u}|y)$.

$$
\begin{aligned}
\mathbb{KL}[q(\mathbf{f}, \mathbf{u})||p(\mathbf{f}, \mathbf{u}|y)] &= \int\int q(\mathbf{f}, \mathbf{u})\log \frac{q(\mathbf{f}, \mathbf{u})}{p(\mathbf{f}, \mathbf{u}|y)} \\
&= \int\int q(\mathbf{f}, \mathbf{u})\log \frac{q(\mathbf{f}, \mathbf{u})p(y)}{p(y|\mathbf{f}, \mathbf{u})p(\mathbf{f}, \mathbf{u})}\mathrm{d}\mathbf{u}\mathrm{d}\mathbf{f} \\
&= \int\int q(\mathbf{f}, \mathbf{u})[\log \frac{1}{p(y|\mathbf{f}, \mathbf{u})} + \log \frac{q(\mathbf{f}, \mathbf{u})p(y)}{p(\mathbf{f}, \mathbf{u})} + \log p(y)]\mathrm{d}\mathbf{u}\mathrm{d}\mathbf{f} \\
&= -\int\int q(\mathbf{f}, \mathbf{u})[\log p(y|\mathbf{f}, \mathbf{u}) - \log \frac{q(\mathbf{f}, \mathbf{u})p(y)}{p(\mathbf{f}, \mathbf{u})}]\mathrm{d}\mathbf{u}\mathrm{d}\mathbf{f} + \log p(y) \\
&= -\mathrm{ELBO} + \log p(y)\,.
\end{aligned}
\tag{2}
$$

Since $\mathbb{KL}[q(\mathbf{f}, \mathbf{u})||p(\mathbf{f}, \mathbf{u}|y)] \geq 0$, Eqn (2) shows that ELBO is a lower bound on the log marginal likelihood $\log p(y)$. In addition, since $\log p(y)$ is independent from $q(\mathbf{f}, \mathbf{u})$, maximizing the ELBO will automatically minimize $\mathbb{KL}[q(\mathbf{f}, \mathbf{u})||p(\mathbf{f}, \mathbf{u}|y)]$.

**Marginal Variational Posterior with Whitening.** In our model, we apply the "whitening" operation proposed in [19]. Specifically, we first define $\mathbf{u} = \mathbf{L}\mathbf{v}$, where $\mathbf{L}\mathbf{L}^T = K_{ZZ}$ and $p(\mathbf{v}) = \mathcal{N}(\mathbf{0}, \mathbf{I})$. Instead of directly defining $q(\mathbf{u})$, here, we define a variational distribution for $\mathbf{v}$, denoted as $q(\mathbf{v}) = \mathcal{N}(\mu_v, \mathbf{S})$. Then, we can compute $q(\mathbf{u}) = \mathcal{N}(L\mu_v, L\mathbf{S}L^T)$. Recall that $q(\mathbf{f}, \mathbf{u}) = p(\mathbf{f}|\mathbf{u})q(\mathbf{u})$ and $p(\mathbf{f}|\mathbf{u}) = \mathcal{N}(K_{XZ}K_{ZZ}^{-1}\mathbf{u}, K_{XX} - K_{XZ}K_{ZZ}^{-1}K_{XZ}^T)$, we can compute $q(\mathbf{f})$ as

$$q(\mathbf{f}) = \int p(\mathbf{f}|\mathbf{u})q(\mathbf{u})\mathrm{d}\mathbf{u} = \mathcal{N}(K_{XZ}K_{ZZ}^{-1/2}\mu_v, K_{XX} + K_{XZ}K_{ZZ}^{-1/2}(\mathbf{S} - \mathbf{I})K_{ZZ}^{-1/2}K_{XZ}^T). \tag{3}$$

Below, we denote $\mu_f = K_{XZ}K_{ZZ}^{-1/2}\mu_v$ and $\Sigma_f = K_{XX} + K_{XZ}K_{ZZ}^{-1/2}(\mathbf{S} - \mathbf{I})K_{ZZ}^{-1/2}K_{XZ}^T$. Note that, here we use the true marginal variational posterior, in our implementation, we also enable the widely applied SoR approximation [23], i.e., $q(\mathbf{f}|\mathbf{u}) \approx K_{XZ}K_{ZZ}^{-1/2}\mu_v$. With SoR, $q(\mathbf{f}) \approx \mathcal{N}(\mu_f, K_{XZ}K_{ZZ}^{-1/2}\mathbf{S}K_{ZZ}^{-1/2}K_{XZ}^T)$. It should also be noted that, with whitening, the variational parameters change from $\{\mu, \Sigma\}$ to $\{\mu_v, \mathbf{S}\}$ and the KL divergence term in ELBO becomes $\mathbb{KL}[q(\mathbf{v})||p(\mathbf{v})]$.

**Expected Conditional Log Likelihood in Regression Model.** Recall that our regression model has an analytical form of the likelihood term in the ELBO. Here, we derive this analytical from of the expected conditional log likelihood. Specifically, we first rewrite our regression model as follows:

$$\mathbf{f}|\mathbf{X} \sim \mathcal{N}(\mathbf{0}, k = \alpha_t^2 k_{\gamma_t} + \alpha_e^2 k_{\gamma_e}), \quad y_i|\mathbf{f}^{(i)} \sim \mathcal{N}(\mathbf{f}^{(i)}\mathbf{w}^T, \sigma^2). \tag{4}$$

With the $q(\mathbf{f})$ in Eqn. (3), we then compute the expected conditional log likelihood as

$$\begin{aligned}
\mathbb{E}_{q(\mathbf{f})}[\log p(y|\mathbf{f})] = \mathbb{E}_{q(\mathbf{f})}[\log p(y|\mathbf{f})] &= \mathbb{E}_{q(\mathbf{f})}[\frac{-N}{2}[\log \sigma^2 + \log 2\pi + \frac{1}{\sigma^2}(\mathbf{y} - \mathbf{F}\mathbf{w}^T)^T(\mathbf{y} - \mathbf{F}\mathbf{w}^T)]] \\
&= \frac{-N}{2}[\log \sigma^2 + \log 2\pi + \frac{1}{\sigma^2}\mathbb{E}_{q(\mathbf{f})}[(\mathbf{y} - \mathbf{F}\mathbf{w}^T)^T(\mathbf{y} - \mathbf{F}\mathbf{w}^T)]],
\end{aligned} \tag{5}$$

where $\mathbb{E}_{q(\mathbf{f})}[(\mathbf{y} - \mathbf{F}\mathbf{w}^T)^T(\mathbf{y} - \mathbf{F}\mathbf{w}^T)]$ can be computed as follows:

$$\begin{aligned}
\mathbb{E}_{q(\mathbf{f})}[(\mathbf{y} - \mathbf{F}\mathbf{w}^T)^T(\mathbf{y} - \mathbf{F}\mathbf{w}^T)] &= \mathbb{E}_{q(\mathbf{f})}[\mathbf{y}^T\mathbf{y} - \mathbf{w}\mathbf{F}^T\mathbf{y} - \mathbf{y}^T\mathbf{F}\mathbf{w}^T + \mathbf{w}\mathbf{F}^T\mathbf{F}\mathbf{w}^T] \\
&= \mathbf{y}^T\mathbf{y} - \mathbf{w}\nu_f^T\mathbf{y} - \mathbf{y}^T\nu_f\mathbf{w}^T + \mathbf{w}\mathbb{E}_{q(\mathbf{f})}[\mathbf{F}^T\mathbf{F}]\mathbf{w}^T \\
&= \sum_i(y_i^2 - 2\nu_f^{(i)}\mathbf{w}^T) + \mathbf{w}\mathbb{E}_{q(\mathbf{f})}[\mathbf{F}^T\mathbf{F}]\mathbf{w}^T,
\end{aligned} \tag{6}$$

where $\nu_f \in \mathbb{R}^{N \times T}$ is the matrix form of $\mu_f$. $\nu_f^{(i)} \in \mathbb{R}^{1 \times T}$ is the $i$-th row of $\nu_f$, representing the mean of the variational posterior of $\mathbf{F}^{(i)}$. After computing the expectation of each element in $\mathbf{F}^T\mathbf{F}$ and combine them together, we have

$$\mathbb{E}_{q(\mathbf{f})}[\mathbf{F}^T\mathbf{F}] = \sum_N[\Sigma_f^{(i)} + (\nu_f^{(i)})^T\nu_f^{(i)}], \tag{7}$$

where $\Sigma_f^{(i)} \in \mathbb{R}^{T \times T}$ is the covariance matrix of the variational posterior of $\mathbf{F}^{(i)}$. Plugging Eqn. (7) into Eqn. (6) and Eqn. (5), we have

$$\mathbb{E}_{q(\mathbf{f})}[(\mathbf{y} - \mathbf{F}\mathbf{w}^T)^T(\mathbf{y} - \mathbf{F}\mathbf{w}^T)] = \frac{-N}{2}[\log \sigma^2 + \log 2\pi + \frac{1}{\sigma^2}\sum_i((y_i - \nu_f^{(i)}\mathbf{w}^T)^2 + \mathbf{w}\Sigma_f^{(i)}\mathbf{w}^T)]. \tag{8}$$

With the analytical form in Eqn. (8), we can minimize the exact ELBO for our regression model without any approximation.

**Predictive Distributions.** Although our model mainly focuses on providing explanations, it can also perform prediction with the predictive distribution. In the following, we derive the predictive distribution of our regression and classification model with the variational distributions. Given a set

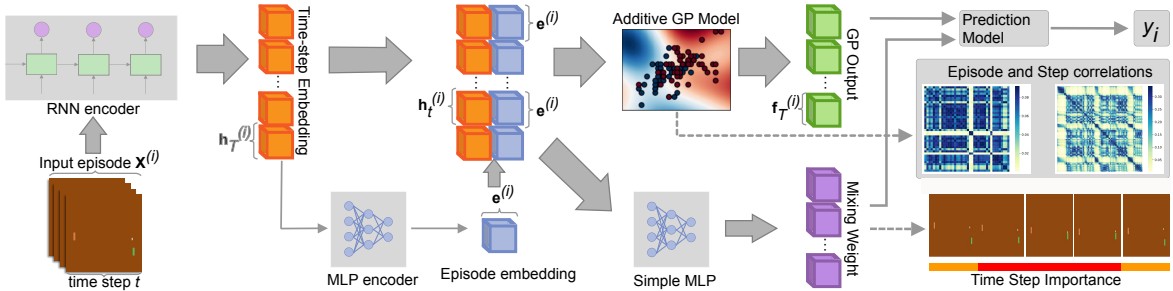

Figure S1: Overview of EDGE with a mixing weight given by an MLP.

of testing episodes $\mathbf{X}_* \in \mathbb{R}^{N_* T \times (d_s + d_a)}$, we first compute the variational posterior of their GP outputs $\mathbf{f}_*$ according to Eqn. (3).

$$q(\mathbf{f}_*) = \int p(\mathbf{f}_*|\mathbf{u})q(\mathbf{u})\mathrm{d}\mathbf{u} = \mathcal{N}(K_{X_*Z}K_{ZZ}^{-1/2}\mu_v, K_{X_*X} + K_{X_*Z}K_{ZZ}^{-1/2}(\mathbf{S}-\mathbf{I})K_{ZZ}^{-1/2}K_{X_*Z}^T), \quad (9)$$

where $\{\mu_v, \mathbf{S}\}$ are the solved variational parameters. Below, we denote the mean and covariance matrix in $q(\mathbf{f}_*)$ as $\mu_*$ and $\Sigma_*$. After obtaining $q(\mathbf{f}_*)$, we then discuss how to conduct prediction in our regression and classification model. Regarding the regression model, which has a Gaussian likelihood, we can directly compute the marginal likelihood distribution as the predictive distribution, i.e.,

$$p(\mathbf{y}) = \int p(\mathbf{y}|\mathbf{f}_*)q(\mathbf{f}_*)\mathrm{d}\mathbf{f}_* = \mathcal{N}(\mu_y, \Sigma_y), \quad (10)$$

where $\mu_y \in \mathbb{R}^{N_*}$ can be computed as

$$\mu_y = \mathbb{E}[\mathbf{y}] = \mathbb{E}[\mathbf{F}_*\mathbf{w}^T] = \nu_*\mathbf{w}^T, \quad (11)$$

where $\nu_* \in \mathbb{R}^{N_* \times T}$ is the matrix form of $\mu_*$. Then ,we compute $\Sigma_y \in \mathbb{R}^{N_* \times N_*}$ as

$$\Sigma_y = \mathrm{Var}[\mathbf{y}] = \mathrm{Cov}[\mathbf{F}_*\mathbf{w}^T, \mathbf{F}_*\mathbf{w}^T] + \mathbf{I}\sigma^2, \quad (12)$$

where $\mathrm{Cov}[(\mathbf{F}_*\mathbf{w}^T)_i, (\mathbf{F}_*\mathbf{w}^T)_j] = \mathbf{w}(\Sigma_*)_{iT:(i+1)T, jT:(j+1)T}\mathbf{w}^T$. After computing the marginal predictive distribution, we can make prediction using its mean $\mu_y$ and access the prediction uncertainty from $\Sigma_y$.

For the classification model, the marginal likelihood is intractable due to the non-Gaussian likelihood. We follow the prediction procedure proposed in [8] and use the MC method to make predictions. Specifically, we first sample $B$ samples from $q(\mathbf{f}_*)$ and compute the conditional likelihood distribution $p(y|\mathbf{f}_*)$ using the drawn samples. Then, we compute the mean of the probability in the conditional distributions (i.e., $\frac{1}{B}\sum_b \mathrm{softmax}(F^{(b)}\mathbf{W}^T)$) as the final predictions.

**EDGE with an Input-specific Mixing Weight.** Recall that our proposed model can provide input-specific explanations by replacing the constant mixing weight with a neural network. As is shown in Fig. S1, we use a simple MLP $e_{\phi_w}$ with three layers, i.e., a linear layer with $T$ number of neurons, a LeakyReLU activation layer, and a linear layer with $TK$ number of neurons. Given the episode encoding of $N$ episodes $\mathbf{C} \in \mathbb{R}^{N \times T \times 2q}$, in which each element is the concatenation of that time step's unique embedding and the episode embedding of the corresponding episode (i.e., $[\mathbf{h}_t^{(i)}, \mathbf{e}^{(i)}]$). We first sum the last dimension of each element and obtain $\mathbf{C}' \in \mathbb{R}^{N \times T}$ (i.e., $\mathbf{C}' = \sum_c \mathbf{C}_{\cdot,\cdot,c}$). Second, we input $\mathbf{C}'$ into the network and get the corresponding output $e_{\phi_w}(\mathbf{C}) \in \mathbb{R}^{N \times TK}$, where $K$ is the total number of classes in our classification model. Third, we transform $e_{\phi_w}(\mathbf{C})$ into the input-dependent mixing weight $\mathbf{W}_x \in \mathbb{R}^{N \times T \times K}$. Finally, we manipulate the GP output $\mathbf{F} \in \mathbb{R}^{N \times T}$ with $\mathbf{W}_x$ and obtain the predictions $\mathbf{P} \in \mathbb{R}^{N \times K}$. To ensure the explainability and stability, we borrow the idea from [1] and design a local-linear regularization for $e_{\phi_w}$. Note that since our feature extractor is non-parametric, the regularization proposed in [1] is not applicable to our model. Specifically, to ensure local linearity, we propose to minimize $\mathcal{L}_e = \|e_{\phi_w}(\mathbf{C}') - e_{\phi_w}(\mathbf{C}' + \epsilon_c)\|_1$ together with the ELBO, where $\epsilon$ is a local

Table S1: Hyper-parameter choices of our method and the baseline approaches in the selected games. The numbers in the bracket of "CNN" represent the number of kernels in each layer. The numbers in the bracket of "Embedding", "MLP", and "GRU" refer to the hidden dimensions.

| Games | Hyper-parameters shared by our method and the baseline approaches | | | | | | | Hyper-parameters unique to our method | |
| | Observation/State encoder | Action encoder | RNN encoder/classifier | Batch size | Epochs | Optimizer | learning rate | Number of inducing points | $\lambda$ |
|---|---|---|---|---|---|---|---|---|---|
| Pong | CNN(32, 32, 32, 16) | Embedding(16) | GRU ($q$ =4) | 40 | 100 | Adam | 0.01 | 100 | 0.1 |
| You-Shall-Not-Pass | MLP(64, 32) | MLP(64, 32) | GRU ($q = 8$) | 40 | 200 | Adam | 0.01 | 600 | 0.01 |
| Kick-And-Defend | MLP(64, 32) | MLP(64, 32) | GRU ($q = 8$) | 40 | 200 | Adam | 0.01 | 600 | 0.01 |
| CartPole | MLP(32, 16) | Embedding(4)→MLP(32, 16) | GRU ($q = 4$) | 40 | 200 | Adam | 0.01 | 100 | 0.01 |
| Pendulum | MLP(32, 16) | MLP(32, 16) | GRU ($q = 4$) | 40 | 100 | Adam | 0.01 | 600 | 0.01 |

random perturbation added to the $\mathbf{C}'$. By minimizing $\mathcal{L}_e$, we can let $e_{\phi_w}$ to be almost a constant around the local area of each input and thus force the prediction model to be local-linear. In this work, we only apply this input-specific mixing weight to the classification model because it will make the exact computation of the expected log-likelihood in the regression model much more complicated and even intractable.

# S2  Implementation Details and Experiment Setups

## S2.1  Implementations and Hyper-parameter selections

**Implementations.**    We implement `EDGE` using the `pytorch` [22] and the `gpytorch` [8] package. Regarding the baseline approaches, we implemented them based on the codes released in their original paper – Rudder: `https://github.com/ml-jku/rudder`; Input-Cell Attention RNN used in the saliency methods: `https://github.com/ayaabdelsalam91/Input-Cell-Attention`, saliency methods: `https://github.com/PAIR-code/saliency`; Attention: `https://github.com/sarahwie/attention`; Rational Net: `https://github.com/taolei87/rcnn`. A preliminary version of our software system with `EDGE` and all the baseline approaches is attached to the supplementary material.

**Hyper-parameters.**    Table S1 shows the hyper-parameter choices of our experiments. First, we discuss the hyper-parameters that are shared across all the methods – network structures and training hyper-parameters. Regarding network structures, recall that we concatenate the state/observation and action at each step before inputting them to an RNN. More specifically, we apply a state/observation encoder and an action encoder to transform the original states and actions into a hidden representation. Since different games have different forms of states and actions, we use different network architectures for them. In the pong game, state/observation is an image of the current snapshot of the environment, and action is discrete. Here, we use a CNN with 4 layers, in which each layer has the kernel size of 3, the stride size of 2, and the "ReLU" activation function, as the state/observation encoder and an Embedding layer as the action encoder. Regarding the MuJoCo and Pendulum games, both observation/state and action are vectors with continuous values. In these games, we directly concatenate the state and action and input them into an MLP encoder. For the CartPole games, where the state is a continuous vector and action is discrete, we transform the action into a continuous vector using an Embedding layer and input it together with the state into an MLP encoder. With the hidden representations of each time step, we then input it into an RNN with GRU cells except for the RNN in RNN+Saliency (In RNN+Saliency, we follow the original setup in [13] and use LSTM as the RNN cell). We adopt the widely applied "Tanh" attention as the attention layer in our attention+RNN model. For the baseline approaches, we directly use the RNN as the predictor (*i.e.,* Seq2one structure). For our method, as is introduced in Section 3.2, we use the RNN together with a one-layer MLP to derive the time-step embedding and the episode embedding. As for the training hyper-parameters, our method shares the same choices as the baseline approaches except for the learning rate in the Kick-And-Defend game (See Table S1 for detailed values). Second, our method introduces two unique hyper-parameters – number of inducing points ($M$) and $\lambda$, where $\lambda$ represents the coefficient of the lasso regularization added on the mixing weight (To encourage more understandable explanations, we add a lasso regularization on the mixing weight $\mathbf{W}/\mathbf{w}$). Table S1 shows the choice of these two hyper-parameters in each game. In Section S3, we further study the sensitivity of our method against $M$ and $\lambda$.

Table S2: Descriptions of the episode dataset of each game. "Discrete $(X)$" refers to categorical actions with the $X$ possible choices. "Vector $(X)$" refers to continuous state/action vectors with the dimensionality of $X$. "Classification (2)" stands for the classification task with 2 possible classes. To enable batch operation, we pad all the episodes in the same game to the same length $T$.

| Games | Observation/state | Action | $T$ | Training size | Testing size | Task type |
|---|---|---|---|---|---|---|
| Pong | Image (80, 80, 1) | Discrete (6) | 200 | 21500 | 1880 | Classification (2) |
| You-Shall-Not-Pass | Vector (380) | Vector (17) | 200 | 31900 | 2000 | Classification (2) |
| Kick-And-Defend | Vector (380) | Vector (17) | 200 | 31500 | 2000 | Classification (2) |
| CartPole | Vector (4) | Discrete (2) | 200 | 29500 | 4200 | Regression |
| Pendulum | Vector (3) | Vector (1) | 100 | 28000 | 4000 | Regression |

## S2.2 Experiment Setups

**Selected games, agents, and episodes.** In our experiments, we select three games with delayed rewards – Atari pong, MuJoCo You-Shall-Not-Pass, and MuJoCo Kick-And-Defend, and two games with instant rewards – OpenAI gym CartPole and Pendulum. For the descriptions of the Atari and OpenAI gym games, readers could refer to [21, 5]. These three games are single-player games, and we directly use the agent in each game as our target agent. Regarding MuJoCo games, readers could find the introductions of their game environments and reward designs in [3]. Note that these games are two-player games, we select the runner in You-Shall-Not-Pass and kicker in Kick-And-Defend as our target agent. Section 4 mentioned that we download a well-trained policy for each game. Specifically, we download the policy in the Pong game from `https://github.com/greydanus/baby-a3c` and the policies in the MuJoCo games from `https://github.com/openai/multiagent-competition`. For the CartPole and Pendulum game, we get the agent from `https://github.com/DLR-RM/rl-trained-agents/tree/d81fcd61cef4599564c859297ea68bacf677db6b/ppo`. All the agents are trained with policy gradient methods (A3C for the pong policy and PPO for the MuJoCO and OpenAI Gym policies). After obtaining the target agents, we run each agent in the corresponding environment and collect a set of training and testing episodes by varying the random seed. Table S2 shows the descriptions of these episode datasets including training-testing split, state-action dimensions, and episode length. Table S2 also shows the type of prediction task on each game. Regarding the games with delayed rewards, we conduct the classification with 2 possible classes (*i.e.,* the target agent wins or losses a game). For the games with instant rewards, we conduct the regression task. In our experiments, we use the training set to train the explanation models and the testing set for evaluations and use cases. Supplementary material includes the trained explanation models used in evaluations and use cases.

**Computational resources.** In our experiments, we use a server with 4 NVIDIA RTX A6000 GPUs to train and test the explanation models.

**Other Related Issues.** As is introduced above, the existing assets used in our experiments are the pretrained agents. We checked the GitHub repositories, from where we download them, and do not find the license information for the Pong and MuJoCO policies. The license for the CartPole and Pendulum is the MIT License. Since all the agents are just neural networks rather than actual data, they do not contains personally identifiable information or offensive content.

# S3 Additional Evaluations on Games with Delayed Rewards

Recall that, besides evaluating the explanation fidelity with regard to the original RL agent and environment, we also compare our method with the baselines approaches from the following dimensions: reward prediction performance, explainability, stability, and efficiency. In this section, we introduce the designs of these experiments on the games with delayed rewards and discuss the corresponding results. Section S5 will present these evaluations on the games with instant rewards. Note that we apply the input-specific mixing weight to our classification model, in this section, we report the results of our method with a constant mixing weight (denoted as `EDGE`) and that with an input-specific mixing weight (denoted as `EDGE_x`).

Table S3: The testing accuracy of each method on the selected games. "`EDGE_x`" refers to our model with an input-specific mixing weight. Note that "Rudder" is designed only for regression tasks, we use it to directly predict the value of $y_i$ and report the MAE (*i.e.*, $\frac{1}{N}\sum_i |\hat{y}_i - y_i|$).

| Games | Rudder | Saliency (%) | Attention (%) | RatNet (%) | `EDGE` (%) | `EDGE_x` (%) |
|---|---|---|---|---|---|---|
| Pong | 0.012 | 88.9 | 99.9 | 97.3 | 99.9 | 99.9 |
| You-Shall-Not-Pass | 0.018 | 99.2 | 99.1 | 99.3 | 99.2 | 99.1 |
| Kick-And-Defend | 0.011 | 98.3 | 98.3 | 98.8 | 99.7 | 98.9 |

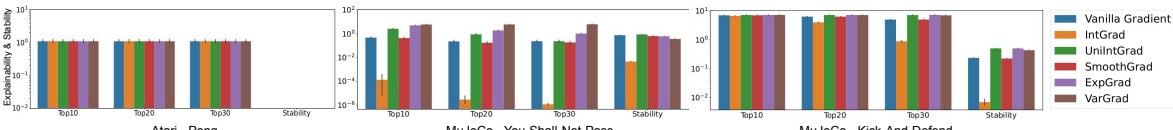

(a) Mean and standard error of the explanability and stability scores obtained by each saliency method. "IntGrad" refers to the integrated gradient and "UniIntGrad" stands to the integrated gradient with uniform baseline. The stability of each saliency method on the Pong game is 0.

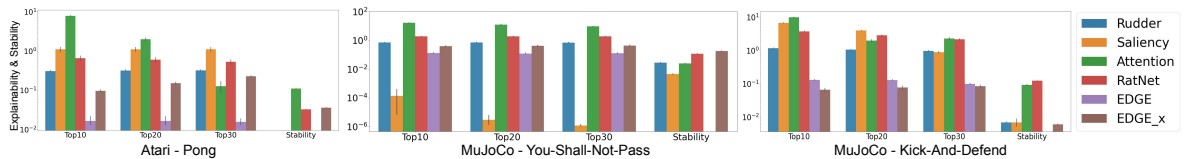

(b) Mean and standard error of the explanability and stability scores obtained by `EDGE` and the baseline approaches. The stability of `EDGE` on all the games is 0. The stability of Rudder and Saliency method on the Pong game is also 0.

Figure S2: Explainability and stability comparison across the selected explanation methods.

**Model Performance.** To evaluate how well each method could predict the final rewards, we test the prediction model in each method on the testing episodes and record the testing accuracy of each game in Table S3. As we can first observe from the table, all the methods could obtain a decent prediction performance except for the saliency method on the Pong game. We take a closer look at this case and find that the RNN method in the saliency method completely biases towards the winning episodes on the Pong game and cannot recognize the losing ones. Table S3 also shows that overall `EDGE` establishes the highest testing accuracy. This result confirms the benefits of our GP feature extractor (*i.e.*, capturing the time step and episode correlations). Finally, we observe that the input-specific mixing weight slightly reduces the testing accuracy of our proposed method. We speculate that this is because the additional model capacity introduced by the local linear MLP causes overfitting. This overfitting problem can be mitigated by increasing the regularization strength.

**Explainability and Stability.** After evaluating the performances of the prediction models, we then evaluate the explainability and the stability of each selected explanation method. Specifically, explainability represents how well an explanation method could explain the prediction model. In our evaluation, we use the metric proposed in [7, 12, 27] to quantify the explainability. Formally, given a normalized explanation $\mathbf{E}_i \in \mathbb{R}^T$ of an input episode $\mathbf{X}_i$, we define the explainability metric as $-\log F^{c_i}(\mathbf{E}_i \odot \mathbf{X}_i)$. Here $\mathbf{E}_i \odot \mathbf{X}_i$ represents multiplying each entity (*i.e.*, the state and action at each step) in $\mathbf{X}_i$ with the corresponding element in $\mathbf{E}_i$, encoding the overlap between the object of interest and the concentration of the explanation. $F^{c_i}(\cdot)$ refers to the model prediction of the true class of $\mathbf{X}_i$. By viewing the explanation as weights of input entities, a faithful explanation should weight important entities more highly than less important ones and thus give rise to a higher predicted class score and a lower metric value. Note that we apply the top-$K$ normalization, i.e., setting the value of the top-$K$ important time steps as 1 and the rest time steps as 0. We do not use the 0-1 normalization because multiplying it with discrete actions will result in invalid actions. It should also be noted that explainability evaluates the faithfulness of the explanation with regard to the prediction

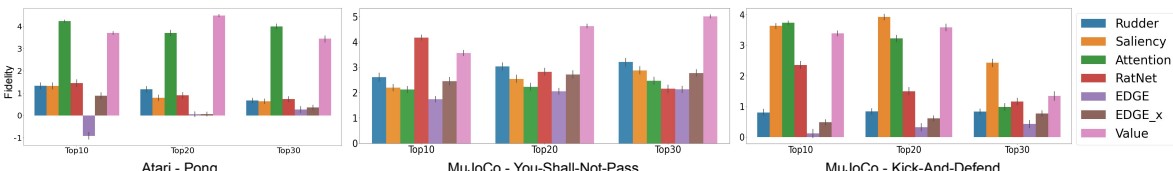

Figure S3: Mean and standard error of the fidelity scores obtained by our method, selected baselines, and the value function. Note that we use the saliency method demonstrates the highest explainability (*i.e.,* integrated gradient) in this experiment (See Fig. S2(a)).

model. We do not need to keep the perturbations (*i.e.,* $\mathbf{E}_i \odot \mathbf{X}_i$) to be physically realistic. In our experiments, we set $K = 10/20/30$. Fig. S2(b) shows the explanability comparisons between EDGE and the baseline approaches. First, we observe that neither Attention nor Rational Net can consistently outperform the post-training explanation approaches (*i.e.,* saliency methods and Rudder). This result confirm the first limitation of the existing self-explainable methods discussed in Section 3.2, i.e., adding the explanation module in front of the prediction model cannot faithfully explain the associations learned by the prediction model. In comparison, with a different explanation module design, EDGE is able to outperform the baseline approaches in most setups. This result verifies the effectiveness of our explanation module design. This result also confirms that by capturing the unique correlations exhibited in the RL episodes, our method could better fit these episodes and thus give rise to higher explainability. We notice a corner case on the You-Shall-Not-Pass game where the saliency method shows a higher explainability than our method. As part of future work, we will investigate the reasons behind this result. Our future work will also explore the reasons behind the different performance of EDGE and EDGE_x on these games. Note that Fig. S2(a) shows the comparison between the six selected saliency methods. Overall, the integrated gradient demonstrates the highest explainability on the selected games. We also observe that all the saliency methods show a similar performance on the Pong games. We speculate this is caused by the model bias mentioned above.

In addition to explainability, we also evaluate the stability of each explanation method against random perturbations added to the input. Specifically, we use the following metric [27] to evaluate the stability: $\mathbb{E}_{\epsilon_s \sim \mathcal{N}(0,\sigma_s^s)} \frac{\|E(\mathbf{X},F^c)-E(\mathbf{X}+\epsilon_s,F^c)\|_2}{\|\epsilon_s\|_2}$, where $\sigma_s^s$ controls the perturbation strength. In our experiments, we set it as 0.05 times of the maximum value range of input features. A lower metric value represents a more stable explanation. Here, we use the MC method to estimate the expectation, i.e., sampling $\epsilon_s$ 10 times and computing the mean of the 10 metric values obtained from the sampled $\epsilon_s$. Note that to ensure the legitimacy of the perturbation, we only add $\epsilon_s$ to the states and actions with continuous values. Fig. S2 shows the results of our methods and all the comparison baselines. As shown in the figure, EDGE demonstrates the lowest the metric value (*i.e.,* 0). This is because the explanation of EDGE is a global explanation, which is robust to input perturbation. In comparison, replacing the constant mixing weight with a neural network jeopardizes the explanation stability. We can further improve the stability of EDGE_x by increasing the regularization strength or training set size.

**Fidelity.** Section 4 shows the fidelity comparison between EDGE and the selected baseline approaches. Here, we further show the fidelity of two other methods – EDGE_x and the explanations drawn from the value function. Regarding the second method, we use the value function of the target agent and treat the time steps with the top-$K$ value function outputs as the top-$K$ important features. Fig. S3 shows the comparison across all the methods. We first observe that the value function cannot faithfully reflect the associations between input episodes and the final rewards. We believe there are two reasons behind this result. First, the policy training inevitably introduces errors to the value function approximation, resulting in inaccurate explanations. More importantly, value function expresses the expected total return of a state, rather than the contribution of the current state to a game's final reward. In other words, it is not designed to capture the specific associations between episodes and their final rewards. Consequentially, the explanations drawn from the value function cannot faithfully represent the above associations. Note that since all of the target agent are trained by policy gradient, we use their

Table S4: Training/explanation time of each method on the selected games. Regarding the saliency method, we record the explanation time of the integrated gradient method, which demonstrates the highest explanability (See Fig. S2(a)).

| Games | Rudder | Saliency | Attention | RatNet | EDGE | EDGE_x |
|---|---|---|---|---|---|---|
| Pong | 7:05min/0.002s | 16:15min/0.02s | 6:49min/0.002s | 7:26min/0.002s | 7:20min/0.002s | 07:42min/0.007s |
| You-Shall-Not-Pass | 2:49min/0.0002s | 4:41min/0.172s | 2:43min/0.0003s | 2:50min/0.0002s | 3:40min/0.0002s | 03:50min/0.011s |
| Kick-And-Defend | 1:57min/0.0001s | 2:54min/0.132s | 1:55min/0.0001s | 2:02min/0.0002s | 3:08min/0.0002s | 03:31min/0.009s |

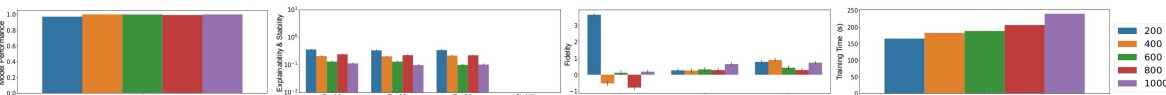

(a) Evaluation performance comparison between the models trained with different number of inducing points.

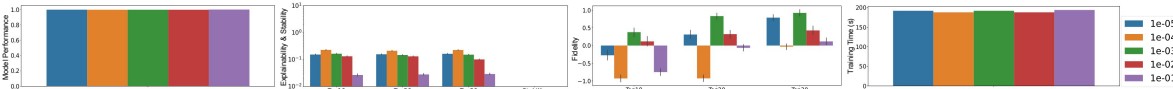

(b) Evaluation performance comparison between the models trained with different $\lambda$.

Figure S4: Hyper-parameter sensitivity test results of EDGE on the Kick-And-Defend game. EDGE refers to our model with a constant mixing weight.

value networks in our evaluation. Our future work will evaluate the effectiveness of using Q network (function) to derive explanation. Second, Fig. S3 also shows that EDGE_x has a slight worse fidelity than EDGE . We speculate this is caused by its worse model performance and lower explanability. Finally, we notice that the fidelity in Fig. S3 is not strictly aligned with the explanability in Fig. S2. In other words, there are some cases where an explanation with a high explanability cannot achieve a high fidelity to the RL agent. As part of future works, we will take a closer look into these corner cases and investigate the reasons behind their results.

**Efficiency.** Table S4 shows the training/explanation time of each method. For training, we record the run time of one training epoch. Regarding explanation, we record the run time of deriving one explanation. As we can first observe from the Table, our methods (*i.e.,* EDGE and EDGE_x) introduce only a slight training overhead compared to the baseline approaches. This confirms the efficiency of our parameter learning method. In addition, since our explanations can be directly drawn from the mixing weight, the explanation process takes negligible time. Similarly, the explanation process of all the other methods is also very fast, except for the integrated gradient method, which is an ensemble method requiring multiple gradient computations in one explanation.

**Hyper-parameter Sensitivity.** Finally, we test the sensitivity of our model performance against the different choices of the unique hyper-parameters introduced by our method (*i.e.,* number of inducing points - $M$ and lasso regularization coefficient - $\lambda$). Specifically, we first fix $\lambda = 0.01$ and vary $M = 200/400/600/800/1000$. For each choice of $M$, we train our explanation model, run the above evaluations, and record the corresponding results in Fig. S4(a). We can roughly observe two trends from the figure – (1) the model performance, explainability, and fidelity get better as $M$ increases; (2) the training time becomes longer as $M$ increases. These trends reflect the general model performance and training efficiency trade-off introduced by inducing points, i.e., using more inducing points will improve the model performance but reduce the training efficiency. Despite the existence of this trade-off, Fig. S4(a) also demonstrates that the model is already able to achieve a decent performance with $M = 600$ and the training time of $M = 600$ is acceptable (introducing about 13.9% extra training time compared to $M = 200$). This result matches with the finding in [25]. That is, a small number of inducing points is enough for decent model performance, and the training time increases slowly in a range of $M$. Overall, this experiment shows that by choosing $M$ within a reasonable range, our method could achieve a superior performance without introducing too much overhead compared to the baseline approaches. This property escalates the practicability of our method in that users do not need to

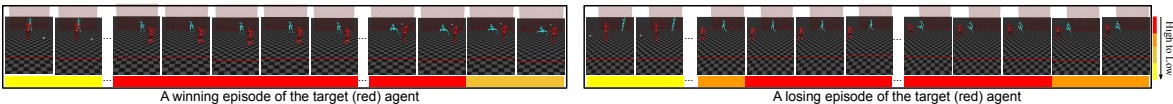

Figure S5: Time step importance of the target agent in the Kick-And-Defend game.

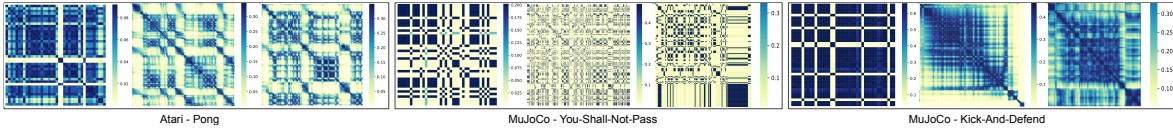

Figure S6: Illustrations of the time step correlations and episode correlations extracted by `EDGE`. For each game, we show the correlations between 40 episodes (left figure), the time step correlations of a winning episode (middle figure), and the time step correlations of a losing episode (right figure).

exhaustively search for the optimal hyper-parameter choices to achieve a decent performance. Second, we fix $M = 600$ and vary $\lambda = 0.1/0.01/0.001/0.0001/0.00001$. Fig. S4(b) shows the results of each model. As we can first observe from the figure, different choices of $\lambda$ have a minor influence upon the model performance and the training time. Fig. S4(b) also shows a general trend that the explainability and fidelity get higher as $\lambda$ increases. This result confirms the benefits of lasso regularization to variable selection in our model. The results of varying $\lambda$ suggest that users could choose a relatively large regularization strength when applying our method.

# S4 Additional Use Cases on Games with Delayed Rewards

In this section, we show more experiment results and discussions of the use cases introduced in Section 5. Note that, following Section 5, we use `EDGE` with a constant mixing weight in the use cases.

**Understanding agent behaviour.** Section 5 visualizes the time steps importance obtained by our method in the Pong and You-Shall-Not-Pass game. Here, we show the explanations in the Kick-And-Defend game and the correlations extracted by our method. Fig. S5 shows the game episodes and the time step importance extracted by our method. Similar to the Pong and You-Shall-Not-Pass game, `EDGE` provides human-understandable explanations in the Kick-And-Defend game. For example, in the winning game, our explanation highlights the time steps when the kicker shoots the ball. This explanation indicates that the kicker (red agent) wins because it shoots a difficult ball that the keeper fails to defend. Fig. S6 visualizes the episode correlations $K_e(\mathbf{X}, \mathbf{X})$ and time step correlations $K_t(\mathbf{X}, \mathbf{X})$ in the selected games. As we can first observe from the figure, the episode correlations demonstrate clear cluster structures. More specifically, in the Pong and Kick-And-Defend game, the episodes form two clusters. After checking the episodes in each cluster, we surprisingly find that the episodes in the same cluster are all winning/losing episodes. In other words, our method successfully groups the games with the same result into one cluster. In the You-Shall-Not-Pass game, despite the episodes have more than two clusters, we also find that each cluster is very pure in terms of the game results with only a few outliers. This result further validates the effectiveness of our method in modeling the joint efforts between episodes. Second, we can also observe that the winning and losing episodes have different time step correlations. Specifically, the losing episodes tend to have more concentrated correlations at the last few steps, whether the time step correlations of the winning episodes are more scattered. As is discussed in Section 3.2, these time step correlations can be used to generated episode-specific explanations from the global ones.

**Launching Adversarial Attacks.** As is discussed in Section 5, we also experiment with the influence of the number of commonly critical steps $K$ upon the attack performances. Specifically, we set $K = 10/20/30$ and record the corresponding attack results of each explanation method in Table S5. Table S5 first shows that the exploitability of each method improves as perturbing more actions (*i.e.,* enlarging $K$). We also observe that overall `EDGE` triggers the most significant performance

Table S5: The changes in target agent's win rate before/after attacks with different choices of $K$. We ran each experiment three times and report the mean and standard error. Section S6 further shows the result of a hypothesis test.

| Games | $K$ | Rudder | Saliency | Attention | RatNet | Our |
|---|---|---|---|---|---|---|
| Pong | 10 | -4.33 (5.13) | -12.33 (2.62) | -5.60 (1.18) | -13.20 (1.88) | **-61.87 (3.92)** |
| | 20 | -5.60 (1.64) | -22.13 (1.23) | -16.27 (1.27) | -23.20 (4.33) | **-64.00 (2.45)** |
| | 30 | -19.93 (4.43) | -30.33 (0.47) | -25.27 (1.79) | -29.20 (4.24) | **-65.47 (2.90)** |
| You-Shall-Not-Pass | 10 | **-9.73 (3.80)** | -6.40 (1.78) | -7.60 (1.77) | -3.54 (2.04) | -9.60 (2.95) |
| | 20 | -21.47 (5.28) | -21.53 (3.49) | -24.27 (6.67) | -13.40 (5.39) | **-29.33 (6.55)** |
| | 30 | -32.53 (4.72) | -29.33 (3.68) | -33.93 (5.77) | -30.00 (1.63) | **-35.13 (2.29)** |
| Kick-And-Defend | 10 | -6.60 (2.95) | -6.00 (6.56) | -5.93 (2.00) | -2.40 (3.08) | **-8.67 (4.73)** |
| | 20 | -14.13 (0.81) | -26.00 (3.46) | -27.13 (6.02) | -4.20 (1.59) | **-33.40 (7.53)** |
| | 30 | -21.80 (3.70) | -37.87 (6.31) | -41.20 (4.70) | -7.13 (2.50) | **-43.47 (4.01)** |

Table S6: Additional results of our patch methods. The upper table shows the patching results of varying the mixing probability $P$ on the Pong game. The low tables shows the patching result of changing the exploration budget $B$ on three games. We ran each experiment three times and report the mean and standard error. Section S6 further shows the result of a hypothesis test.

| Games | Setups | Rudder | Saliency | Attention | RatNet | EDGE |
|---|---|---|---|---|---|---|
| Pong | $P=0.14$ | +0.68 (0.39) | 0.08 (0.06) | +0.13 (0.12) | -1.77 (0.53) | **+1.28 (0.24)** |
| | $P=1$ | +1.89 (1.25) | -1.13 (0.96) | -0.58 (1.81) | -3.66 (1.35) | **+2.75 (0.65)** |
| Pong | $B=10$ | +1.89 (1.25) | -1.13 (0.96) | -0.58 (1.81) | -3.66 (1.35) | **+2.75 (0.65)** |
| | $B=20$ | +4.48 (0.72) | -1.91 (1.40) | -4.00 (1.06) | -2.58 (3.61) | **+4.84 (1.91)** |
| | $B=30$ | **+3.28 (0.88)** | -2.38 (1.74) | -1.23 (1.00) | -5.50 (0.84) | +0.80 (0.57) |
| You-Shall-Not-Pass | $B=10$ | +1.76 (0.17) | +0.92 (0.32) | +0.44 (0.06) | +1.68 (0.50) | **+2.91 (0.32)** |
| | $B=20$ | +1.66 (0.15) | +0.47 (0.12) | +0.31 (0.15) | +1.56 (0.25) | **+3.01 (0.34)** |
| | $B=30$ | +1.34 (0.24) | +0.13 (0.11) | +0.20 (0.06) | +1.42 (0.09) | **+2.79 (0.18)** |
| Kick-And-Defend | $B=10$ | +0.96 (0.1) | +1.17 (0.17) | +0.57 (0.04) | **+1.21 (0.16)** | **+1.21 (0.13)** |
| | $B=20$ | +3.16 (0.49) | +3.21 (0.18) | +2.09 (0.06) | +2.43 (0.39) | **+4.02 (0.31)** |
| | $B=30$ | +3.11 (0.28) | +2.90 (0.30) | +1.84 (0.26) | +3.57 (0.32) | **+3.92 (0.65)** |

drop in all the setups except one case. We defer to future work to study the reason behind that case. Besides summarizing the most critical time steps, an attacker could also record the states of the critical time steps and launch attacks at the most important states (if the total number of the state is within a reasonable range). Our future work will compare the exploitability between the attack based on time steps and the attack based on states. Future works could also explore combining the explanation with the existing adversarial attacks (*e.g.,* manipulating the target agent's observation only at the critical time steps identified by the explanation methods).

**Patching Policy Errors.** As shown in Table 1 in Section 5, our patching method jeopardizes the agent's winning rate on the Pong game when using Attention and Rationale Net as the explanation method. This motivates us to explore a probabilistic mixing policy. That is, when the current state is in the look-up table, the agent chooses the corresponding action in the table with a probability $P$ and the action given by its original policy with the probability $1 - P$. To decide the value of $P$, we run the agent's original policy in the Pong environment for $N_a$ games and record the number of losing games that encounter the states stored in the look-up table (*i.e.,* $N_l$). We compute $P = N_l/N_a$, representing the probability of the agent running into the states in the look-up table and eventually losing the corresponding game. In our experiment, we set $N_a = 500$ and compute $P = 0.14$. We use this probability to rerun the patching method on the Pong game and record the changes in the agent's winning rate before/after patching in the upper table of Table S6. As we can observe from the table, a lower mixing probability indeed alleviates the false positive introduced by Attention and Rationale Net. At the same time, it also decreases the effectiveness of the other explanation methods. This result indicates that users could start with a conservative patching policy by setting a small value for $P$ and

Table S7: The testing MAE of each method on games with instant rewards.

| Games | Rudder | Saliency | Attention | RatNet | EDGE |
|---|---|---|---|---|---|
| Cartpole | 0.01 | 0.03 | 0.006 | 0.035 | 7e-05 |
| Pendulum | 0.006 | 0.008 | 0.005 | 0.03 | 0.009 |

increase the probability if aiming for a better patching performance. Besides the probabilistic mixing policy, we also study the influence of look-up table size on the patching performance. The look-up table size is decided by the exploration budget $B$ and the number of continuous critical steps. In this experiment, we vary the look-up table size by setting different $B$. Specifically, we set $B = 10/20/30$ and report the corresponding patching performance in the lower table of Table S6. Overall, we observe that the patch performance improves as $B$ increases from 10 to 20 and drops as $B$ reaches 30. This result indicates that as the look-up table size increases, it includes more losing games and their corresponding remediation policies. As a result, the patched policy is able to correct more errors and thus achieve a higher winning rate. Oppositely, adding more states into the look-up table will also introduce more new errors. This is because a state may occur both in a winning game and a losing game, and changing the actions in an original winning game may, unfortunately, result in a loss. When the table size reaches a certain point (30 in our experiment), the new errors start to dominate the patched episodes, causing a winning rate drop. This result implies that the users may need to search for an optimal choice of $B$ to get the highest patching performance. As part of future work, we will study the corner cases in this experiment and explore how to search for an optimal look-up table size more systemically and efficiently. Note that the patching results of the saliency method are all zeros on the Pong game because it fails to search for any successful remediation. As a result, the look-up table is empty in those setups. We defer to future work to study the reason behind this result.

Rather than using a mixed policy, we also tried to enhance the original policy via behavior cloning (*i.e.,* fine-tune the policy network with the collected state-action pairs). We found this strategy barely works because the policy network oftentimes "forgets" its learned policy after the behavior cloning. Note that we perform random exploration in our method. If an oracle or a better policy is available, we can perform more efficient exploration by mimicking their actions at the critical steps. Going beyond mixed policy or learning from oracle, a more general patch solution would be to redesign the reward function based on the explanation results (*e.g.,* adding some intermediate rewards to guide the agent taking correct actions at the important steps). As part of future work, we will investigate how to design such intermediate rewards.

# S5    Evaluation on Games with Instant Rewards

In this section, we evaluate our proposed method on two games with instant rewards, i.e., CartPole and Pendulum. Similar to the evaluations on the games with delayed rewards, we also compare EDGE with the baseline approaches from the following five dimensions: model performance, explainability, stability, fidelity, and efficiency. In the following, we introduce the setup of each experiment and discuss the corresponding experiment results. As discussed above, we do not apply the input-specific mixing weight to the regression method in this paper. As a result, we only report the results of our model with a constant mixing weight (denoted as EDGE).

**Model Performance.**    Table S7 shows the testing MAE (*i.e.,* $\frac{1}{N}\sum_i |\hat{y}_i - y_i|$) of the prediction model in each explanation method. Note that since we conduct regression on these games, we use the MAE instead of accuracy as the metric. As we can observe from the table, EDGE demonstrates the lowest MAE on both games, indicating the best prediction performance. This result further confirms the effectiveness of our model design in fitting RL episodes.

**Explainability and Stability.**    In this evaluation, we use a different explainability metric from the classification tasks. Specifically, we define the explainability metric of the regression tasks as $|F(\mathbf{E}_i \odot \mathbf{X}_i) - F(\mathbf{X}_i)|$. This metric is the MAE between the model prediction of the original input and the input weighted by the explanation. A faithful explanation should highlight the important entities

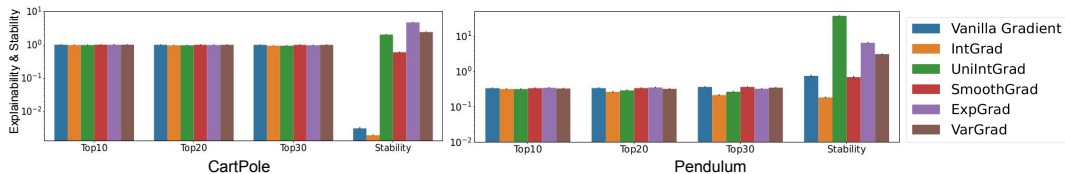

(a) Mean and standard error of the explanability and stability scores obtained by each saliency method.

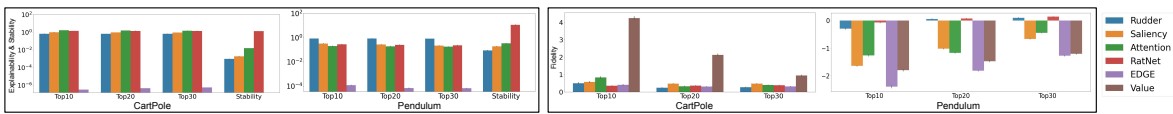

(b) Mean and standard error of the explanability, stability, and fidelity scores obtained by EDGE and the baseline approaches. The stability of EDGE on all the games is 0.

Figure S7: Explanability, stability, and Fidelity comparison across the selected explanation methods.

Table S8: Training/explanation time of each method on the games with instant rewards. Regarding the saliency method, we record the explanation time of the integrated gradient method, which demonstrates the highest explanability (See Fig. S7(a)).

| Games | Rudder | Saliency | Attention | RatNet | EDGE |
|---|---|---|---|---|---|
| Cartpole | 37s/3e-05s | 1:52min/0.02s | 32s/3e-05s | 37s/3e-05s | 1:44min/0.001s |
| Pendulum | 33s/7.6e-05s | 2:24min/0.057s | 35s/7.5e-05s | 37s/6.8e-05s | 1:20min/0.003s |

and thus keep the original prediction value and a lower metric value. Similarly, we also apply the top-$K$ normalization and set $K = 10/20/30$. The left two figures in Fig. S7(b) show the explanability comparisons between EDGE and the baseline approaches. The results have a similar trend as those in Fig. S2(b). This result further demonstrates the superiority of our method in explanability and confirms the benefits of our explanation model design. Fig. S7(a) shows the explanability comparison between the six selected saliency methods. Similar to the results in Fig. S2(a), the integrated gradient also demonstrates the highest explanability on these two games. In this experiment, we apply the same setup and metric as the experiment in Section S3 for the stability comparison. As shown in the left two figures of Fig. S7(b), EDGE demonstrates lowest metric value, indicating the highest stability.

**Fidelity.** We apply the fidelity evaluation method introduced in Section 4 for the fidelity evaluation on the CartPole and Pendulum game. The right two figures in Figure 7(b) show the fidelity comparison between EDGE and the baseline approaches (we also use the integrated gradient as the saliency method). We observe that benefiting from the superior model prediction performance and explanability, our method demonstrates the highest fidelity in most setups. This result further demonstrates the advantage of our self-explainable model over the comparison baselines.

**Efficiency.** Table S8 shows the training/explanation run time comparison between our method and the baseline approaches. Since the games are simpler than the above games with delayed rewards. The run times in Table S8 are smaller than those in Table S4. Similar to the results in Table S4, our method introduces a small training overhead over the existing methods (up to 72s). The explanation times are negligible for all the explanation methods. In summary, with the results in Section S3 and S5, we can safely conclude that with our newly designed self-explainable model and the parameter learning procedure, EDGE improves the baseline approaches from multiple dimensions (*i.e.,* explanability, stability, and fidelity) without introducing too much extra computational cost.

## S6   Hypothesis Test

To further demonstrate the statistical significance of our results, we conducted a paired t-test on the experimental results in Table 1, S5, and S6. Specifically, given a set of results of our method

Table S9: P-value of each experiment in Table 1.

| Applications | Games | Rudder vs. EDGE | Saliency vs. EDGE | Attention vs. EDGE | RatNet vs. EDGE |
|---|---|---|---|---|---|
| Target agent win rate changes before/after attacks | Pong | <0.001 | 0.003 | 0.001 | 0.009 |
| | MuJoCo-Y | 0.19 | 0.13 | 0.35 | 0.04 |
| | MuJoCo-K | 0.02 | 0.08 | 0.28 | 0.003 |
| Target agent win rate changes before/after patching | Pong | 0.29 | 0.02 | 0.09 | <0.001 |
| | MuJoCo-Y | 0.005 | 0.006 | 0.006 | 0.009 |
| | MuJoCo-K | 0.12 | 0.43 | 0.006 | 0.49 |
| Victim agent win rate changes before/after robustifying | MuJoCo-Y | 0.002 | 0.001 | 0.006 | 0.008 |

Table S10: P-value of each experiment in Table S5.

| Games | $K$ | Rudder vs. EDGE | Saliency vs. EDGE | Attention vs. EDGE | RatNet vs. EDGE |
|---|---|---|---|---|---|
| Pong | 10 | <0.001 | <0.001 | <0.001 | <0.001 |
| | 20 | <0.001 | 0.002 | <0.001 | 0.006 |
| | 30 | <0.001 | 0.003 | 0.001 | 0.009 |
| You-Shall-Not-Pass | 10 | 0.51 | 0.12 | 0.21 | 0.05 |
| | 20 | 0.05 | 0.05 | 0.21 | 0.05 |
| | 30 | 0.19 | 0.13 | 0.35 | 0.04 |
| Kick-And-Defend | 10 | 0.21 | 0.36 | 0.12 | 0.15 |
| | 20 | 0.02 | 0.05 | 0.06 | <0.001 |
| | 30 | 0.02 | 0.08 | 0.28 | 0.003 |

($O = \{O_1, O_2, O_3\}$) and that of a baseline ($B = \{B_1, B_2, B_3\}$), we first compute their difference $D = \{O_i - B_i\}$, for $i = 1, 2, 3$. Our non hypothesis for attacks experiments is $H0_1 : \mathbb{E}[D] \geq 0$. The non-hypothesis for policy patch and adversarial defense experiments is $H0_2 : \mathbb{E}[D] \leq 0$. Given these non-hypothesis, we compute the $p$-value for the performance difference of each group of comparison and show the values in Table S9, S10, and S11. For attacks, if $p$ is small, we should reject the $H0_1$, indicating our attack triggers a higher winning rate drop than the comparison baseline and thus has a better exploitability. Regarding the patch and defense experiments, if $p$ is small, we should reject the $H0_2$, indicating our method enables a higher winning rate increase than the comparison baseline and thus has a better performance. Overall, the results in Table S9, S10, and S11 are aligned with those in Table 1, S5, and S6.

# S7 User Study

Recall that Section 2 discusses that previous DRL explanation methods derive interpretations of individual actions by identifying the observation's feature importance regarding the agent's policy network/value function output. Whereas our work highlights the time steps critical to an agent's final result in each episode (*e.g.,* win or loss). In previous research, some researchers conducted user studies to demonstrate the utility of their explanation methods from human perspectives (*e.g.,* [9, 2] uses interpretation to distinguish well-trained (good) and overfitted (bad) agents). In this work, we also performed a user study to demonstrate the utility of our explanation method. Below, we describe the design and results of our user study.

We obtained IRB approval and conducted a user study to compare our proposed explanation method with a representative explanation method [9] that pinpoints the input features essential to the agent's individual actions via a saliency method. Specifically, we first recruited 30 participants with different backgrounds in DRL and DRL explanations (4 participants have published paper(s) in DRL explanation; 6 participants have read some papers about DRL explanation; 10 participants have a general understanding of DRL explanation; 10 participants have never heard of DRL explanations.). Then, we presented an online survey to these participants. This survey aims to compare our explanation method with [9] from two perspectives. (1) How well can the explanations generated from the two approaches help a user to pinpoint a good policy? (2) How well can the explanations help a user perform episode forensics and thus understand why an agent fails or succeeds? We briefly describe the

Table S11: P-value of each experiment in Table S6.

| Games | Setups | Rudder vs. EDGE | Saliency vs. EDGE | Attention vs. EDGE | RatNet vs. EDGE |
|---|---|---|---|---|---|
| Pong | $P$=0.14 | 0.05 | 0.005 | 0.004 | <0.001 |
| | $P$=1 | 0.29 | 0.02 | 0.09 | <0.001 |
| Pong | $B$=10 | 0.29 | 0.02 | 0.09 | <0.001 |
| | $B$=20 | 0.37 | 0.05 | 0.008 | 0.09 |
| | $B$=30 | 0.96 | 0.04 | 0.09 | 0.006 |
| You-Shall-Not-Pass | $B$=10 | 0.005 | 0.006 | 0.006 | 0.009 |
| | $B$=20 | 0.03 | 0.002 | 0.006 | 0.004 |
| | $B$=30 | <0.001 | 0.002 | 0.001 | 0.002 |
| Kick-And-Defend | $B$=10 | 0.12 | 0.43 | 0.006 | 0.49 |
| | $B$=20 | 0.01 | 0.01 | 0.004 | 0.02 |
| | $B$=30 | 0.14 | 0.12 | 0.02 | 0.15 |

design of our user study and the study results below.

(1) Identifying a good policy: Given the representative episodes gathered from two agents of the You-Shall-Not-Pass game (one well-trained and the other overfitted to one specific opponent, i.e., an adversarial agent), we first derived explanations for each of these episodes using the aforementioned interpretation methods. Second, we randomly partitioned the 30 participants into two equally-sized groups and presented the episodes to each group. For Group-A, we also presented the explanations that our method generates. For Group-B, we provided the interpretations that the other method [9] generates. Based on the episodes and their corresponding interpretation, we asked each subject to pinpoint the well-trained agent and asked whether the explanations help identify the good policy. We first discovered that 11 out of 15 participants in Group-A correctly identified the well-trained agent, and 63.6% of them found the explanation helpful. Regarding group-B, 10 out of 15 participants identified the good policy, and 50% of them found the explanation helpful. From the above results, we can get that 7 ($11 \times 0.636$) out of 15 participants in Group-A correctly identified the good policy with the help of our explanations, and 5 ($10 \times 0.5$) out of 15 participants in Group-B correctly identified the good policy with the help of the explanations given by the baseline approach [9]. To compare the ability of two explanation methods in facilitating good policy identification, we conducted a two population proportion test [11]. Specifically, we first set the null hypothesis $H_0$ as $p_1 = p_2$, where $p_1$ and $p_2$ are the probability of correctly identifying the good policy according to our explanations and the explanations given by the baseline approach [9]. Then, we computed the sample probability for each group – Group-A: $\hat{p}_1 = 7/15$; Group-B: $\hat{p}_2 = 5/15$ and the $z$ statistic, i.e., $z = \frac{(\hat{p}_1 - \hat{p}_2)}{\sqrt{\hat{p}_c(1-\hat{p}_c)(\frac{1}{n}+\frac{1}{n})}}$, in which $\hat{p}_c = (7+5)/30$ is the pooled sample proportion and $n = 15$ is the number of participants in each group. Plugging in $\hat{p}_1$ and $\hat{p}_2$, we computed $z = 0.745$. Finally, we computed the percentile $r$ of $z$ in the standard normal distribution and obtained the $p$-value as $2(1 - r) = 0.456$. Since the $p$-value is not that small (e.g., $\leq 0.05$), we fail to reject $H_0$. This result shows that our method demonstrates approximately the same utility as the existing explanations in identifying good/bad policies.

(2) Performing forensics: Given a set of representative episodes gathered from one agent, we used the two above explanation methods to derive explanations for each episode. Then, we present to participants the episodes along with the explanations generated by two different methods. We asked the participants which explanation methods are more beneficial in helping the subject understand why the agent fails/succeeds. We discovered that 21 participants ($70\% = 21/30$) chose our method. This discovery implies that interpreting by highlighting critical time steps could better facilitate episode forensics than explaining by highlighting critical input to the action at each step. We further conducted a binomial test [11] to demonstrate that our explanation method significantly outperforms the existing method [9] in helping policy forensics. In this test, our null hypothesis $H_0$ is $p \leq 0.5$, where $p$ is the probability of choosing our explanation method as more helpful for policy forensics. Then, we computed the percentile $r$ of $Y$ in the binomial distribution $B(30, 0.5)$, where $Y = 21$ is the number of participants that chose our method as the more helpful one. Finally, we compute the $p$-value as $1-r = 0.008$. This small $p$-value means we should reject $H_0$, indicating that our method is significantly better than the baseline [9] in facilitating policy forensics.

Table S12: Mean/Standard error/$P$-value of the attack performance of our method and the action preference attack with different choices of $K$. Note that since the action preference attack is not applicable to the You-Shall-Not-Pass and the Kick-And-Defend games, we only conducted the experiment on the Pong game.

|  | $K$ | Action preference attack | EDGE |
|---|---|---|---|
|  | 10 | -50.34/0.94/0.04 | **-61.87/3.92** |
| Pong | 20 | -61.33/1.25/0.19 | **-64.00/2.45** |
|  | 30 | -62.33/0.94/0.17 | **-65.47/2.90** |

More details about our user study: Survey questions for Group-A: `https://forms.gle/SfUCRC WhZEag47gj9`; Survey questions for Group-B: `https://forms.gle/Kkj4z4wapCTDqXN76`. Questions 1-3 are the same in both surveys. They ask about the participant's background and whether he/she understands the game rule and two types of explanations (We found all 30 participants correctly answer Questions 2&3). Questions 4-7 are about identifying good/bad policies. We present four episodes of the well-trained/overfitted agent together with different explanations to the participants (We count a participant as correctly selecting the good policy only if the participant correctly answered all questions.). Questions 9-10 are about forensic evaluation. They are the same in both surveys, where the participants are presented with the same videos. Each video shows an episode of the same agent together with explanations derived by our method and [9]. The participants are asked to choose which explanation is more helpful. As mentioned above, we found that 21 out of 30 participants chose our explanations in both questions.

## S8 Comparison of Our Attack with An Existing Attack

As mentioned in Section 5, existing research has developed various attacks against DRL policies (*e.g.,* [10, 26, 17]). In this section, we compared our attack with the attack proposed in [17]. This attack manipulates the observations at the selected time steps, whereas our attack changes the actions at the important steps identified by our method. They cannot be directly compared due to the different attack spaces (observation space vs. action space). To enable the comparison, we applied the time step selection method developed in [17] to choose time steps and modify the actions at the selected steps. Specifically, the time step selection method in [17] first computes the action preference at each step as $\pi(s, a_{\max}) - \pi(s, a_{\min})$. Here, $\pi$ is the target policy, which outputs the probability of each action. $a_{\max}$ and $a_{\min}$ refers to the action with highest/lowest probability at the state $s$. Then, it ranks the action preference and selects the time steps with high preference scores to launch its attack. As mentioned above, in our experiment, we rank the action preference scores of states in each episode and randomly change the actions at the states with top-$K$ action preference scores (marked as Action preference attack). We also conducted three groups of experiments and reported the mean/stand deviation/p-value of the paired t-test with $H0_1 : \mathbb{E}[D] \geq 0$ as the null hypothesis. The results are shown in Table S12. We observe from Table S12 that our attack has a stronger exploitability, confirming the advantage of our method in identifying important steps. Note that the time step selection method in [17] cannot be applied to the policy networks that directly output the action rather than the action probability (*e.g.,* the policy networks trained by the PPO algorithm). In our experiment, the policies of the You-Shall-Not-Pass and the Kick-And-Defend games are trained by PPO. As such, [17] cannot be applied to these two games and thus we only compared it with our method on the Pong game, where the agent is trained with the A3C method.

## S9 Potential Social Impact

Any work focusing on general statistical methods, machine learning models included, runs the risk of those methods being used for purposes the authors did not consider. As one of these general purpose tools, EDGE is designed to allow for new modes of understanding and improving Reinforcement Learning

(RL) agents in particular.

Reinforcement learning has recently enjoyed successful application in many areas of computer science. For example, gaming, robotics, natural language processing, and computer vision have all seen advancements from RL [16]. With such wide areas of application, RL methods have naturally extended to several business domains; some examples include business management [15], finance [14], healthcare [6], and education [18]. This is to say, our work focuses on a general purpose tool whose applications cannot be well forecast in advance. For any set of these applications, there will inevitably be subsets considered harmful, and others considered beneficial; furthermore, the stated harms and benefits will vary depending on the individuals surveyed.

In considering the potential benefits and harms that may result from our work, we turn to the many such discussions focusing on other general purpose tools. Recently there has been a wide array of literature published on the fairness, benefits, biases, and harms of machine learning. These works focus on aspects such as the biases and discriminatory behavior present in machine learning models, in addition to mitigation strategies that should be employed in deployment and commercialization [4] [24] [20].

EDGE , as a tool to understand and improve RL methods, can be used to amplify biased or unfair models. Conversely, it can also be used to understand the decision-making process of such systems and enable the construction of strategies to mitigate these harmful factors. We consider this situation analogous to that of open source tools; open source software enables hackers and criminals but simultaneously provides benefits to all of us, including visibility into the methods of those employing such tools. In this way, we also believe it is important to construct tools such as EDGEso that we can further understand and improve RL methods that have come before and after it.