# OpenReview forum: "EDGE: Explaining Deep Reinforcement Learning Policies"
_NeurIPS.cc/2021/Conference — NeurIPS 2021 Poster_

### Official Review · Reviewer_5o96 · 2021-07-14

**Rating:** 6
**Confidence:** 3

**Summary:**

This paper proposes a new pipeline for understanding agent behavior + actions, and identifying which timesteps contributed to the final reward. The key component of this method is the pipeline, which consists of the following:

* An RNN which takes in state-action pairs, and "compresses" the information via its RNN-state at every timestep.
* All RNN-states over all time steps are taken and inserted into a zero-mean Gaussian Process (GP), which consists of two independent GPs, one for correlations between time-steps, and another for correlations between episodes.
* A feature vector $\mathbf{f}$ is then sampled from this GP, and used to classify the reward.

Because a GP is involved, variational inference is needed for training.

The paper then experimentally verifies this result on Atari Pong + Adversarial Mujoco tasks by showing competitive performance on fidelity (i.e. metric based on replacing an intelligent agent with random actions) and visualizing which parts of an episode contribute most to the fidelity score. Finally, the method is also applied to successfully perform adversarial attacks which also beat previous baselines.

**Limitations And Societal Impact:**

Yes.

**Main Review:**

Pros:
* This method is applied to scenarios where access to the agent policy is restricted (or might not even be a neural network), and in offline settings, which makes it broad and useful in many settings, compared to saliency (which is specific to neural networks).
* The experiments seem convincing and strong compared to saliency-based methods, and visualizations confirm the validity of the method.


Cons:
* The entire pipeline is pretty large and complex, and involves many different components. It took quite a while for me to understand the overall picture of what was happening, mainly from staring at Figure 1 and matching with the text. There's also a lot of fine-grained details in Sections 3.2 + 3.3, some of which should be moved into the appendix - is it possible to explain the method at a high level using brief math instead? I think much of the method can be simplified for the average reader to digest.
* The paper requires a somewhat restrictive assumption, which is that in every episode, there are specific timesteps that contributed to the overall reward. I feel like this is only applicable to a limited set of RL environments, such as Pong and Adversarial Mujoco, since they only produce a +1/-1 reward at the very end of the episode. Would this method work for environments that always produce rewards at every timestep?

**Time Spent Reviewing:**

4

---

> ### Author Response · Authors · 2021-08-10
> **Response to Reviewer 5o96.**
>
> We thank Reviewer 5o96 for the constructive and insightful suggestions. Please see below for our response.
>
> Reviewer 5o96 first suggested making the technical part more concise. We will follow the reviewer’s suggestion to revise the paper. More specifically, we will highlight the high-level idea of the proposed framework with an overview paragraph/subsection (with a more direct connection to Fig. 1) and move some of the technical details to the Supplementary material.
>
> Reviewer 5o96 also questioned the applicability of our explanation method to games with instant rewards. In Supplement Section S5, we evaluated our proposed method on two games with instant rewards (i.e., environments that produce rewards at each time step) - CartPole and Pendulum and demonstrated that our method could provide high fidelity evaluations on these environments. In Section 6 lines 372-384, we discuss the generalizability of our method to broader types of games.

---

> > ### Author Response · Authors · 2021-08-25
> > **Follow-up with Reviewer 5o96**
> >
> > Thanks the Reviewer 5o96 again for the insightful comments. Since the discussion phase is about to end, we are writing to kindly ask if the reviewer has any additional comments regarding our response. We are at their disposal for any further questions. In addition, if our response addresses the reviewer's concern, we would like to kindly ask if the reviewer could reconsider their score.

---

> > > ### Comment · Reviewer_5o96 · 2021-08-27
> > > **Response to Authors**
> > >
> > > Hi, after discussions among the reviewers, I will increase my score to 6.
> > >
> > > One of the smaller reasons for this is because the authors pointed out in their supplement that the method also can be quite useful for dense reward environments, which was one of the weakness points I proposed.
> > >
> > > Another reason for the score upgrade is due to its potential impact for the future. It suggests that it may be possible to train an RNN (or a Transformer!) on MDP trajectories, and also add a Gaussian Process on top in order to perform interesting ablation studies. Given the recent research on e.g. Decision Transformers [1] for analyzing offline trajectory data, I think this paper adds value to this sub-area, and it would be interesting to see what may happen if the RNN were replaced with a Transformer.
> > >
> > > I still do think that the paper should be cleaned up better in terms of how they setup their pipeline (and not obfuscated with lots of mathematical notation), as it took me a while to understand what was going on.
> > >
> > > [1] Decision Transformer: Reinforcement Learning via Sequence Modeling

---

> > > > ### Author Response · Authors · 2021-08-27
> > > > **Response to Reviewer 5o96**
> > > >
> > > > We thank Reviewer 5o96 for their reconsideration and its reasoning. We find the description of potential future impact illuminating, and will definitely add it to a description of future work. Using a GP to study the decision-making process of a transformer would prove a very interesting experiment indeed, and is an area we hadn’t considered yet.
> > > >
> > > > We also apologize for any misunderstanding caused by the notation. We will ensure that the overview section in the revision is able to clarify that a bit, and also make sure to review the pipeline description itself.

---

### Official Review · Reviewer_mSsX · 2021-07-16

**Rating:** 6
**Confidence:** 3

**Summary:**

This paper presents a method that uses the correlation between different episodes as well as the correlation between states within an episode to provide explanations on deep RL agents. The explanation module mainly consists of an RNN module and deep gaussian processes. Later in the paper, the authors show that the provided explanations could be used to attack deep RL agents, make them more robust, and repair them.


**Limitations And Societal Impact:**

The authors have adequately addressed the limitations of their proposed method.

**Main Review:**

The idea presented in the paper is interesting, and the approach seems promising. However, there are a few points that need to be addressed in order to increase the quality of the paper. They are mentioned below.

In methods where an attention module is embedded in the deep RL agent's architecture, provided explanations could be easily misleading because the soundness of such approaches is not apparent. The reason for the unclear inferred explanation is that the attention module parameters are affected by the input observations and memory of the agent, which are ignored when an interpretation is given. This misleading explanation makes it hard to find if the attention patterns embed strategic information or just information highlighted by the attention that is key to decisions.

Regarding lines 56-57: The authors claim that their proposed method is the first to explain deep RL policies from a sequence of its past actions and states. However, that is not an accurate claim. [1] and [2] both try to interpret deep RL policies with respect to the actions the agent has taken and the values it has in its memory, resulting in interesting insights into the workings of deep RL agents.

Regarding lines 79-80: How do authors compare their method to [2]? Because in [2], the time steps in which critical decisions are made are also identified and investigated.

Regarding lines 269-271: If the policy has a memory in any form of RNN, this is simply invalid because manipulating the trajectory of $(t_i, ..., t_{i+l})$ would affect the policy's memory.  So simply feeding state at $i+l+1$ would not be sound.

Regarding line 275: How does $d_{max}$ is calculated?

Regarding line 277: How about normalizing the first term, $log(p_l)$, of the equation by a coefficient in $[0,1]$? The reason is that it could have the most effects on the fidelity score, thus undermining the second term, $log(p_d)$.

Regarding Section 5: Many other things could be extracted from these sorts of explanations (lines 298-299 and 301-302). For example, one could conclude that the agent is just focusing on hitting the ball, thus not losing, rather than focusing on hitting it in a special way. We need to be very careful about our conclusions.

Regarding lines 297-298: It is mentioned that "EDGE highlights the time steps when the agent hits the ball as the key steps leading to a win." Is this claim consistent across different policies?

Regarding line 307: It is mentioned that the examples extracted from three games are provided in the Supp material. However, to show the soundness of the approach, a diverse set of games should be investigated.

Regarding Table 1: Please provide variance as well as the mean.

Regarding lines 312-313: How about strategically timed attacks? They do the same without the need for any explanations [3]. A comparison between the two methods could improve the claim in the paper.

Regarding Patching Policy Errors: It is mentioned that a table could be formed from the state-action pairs to be used as a remediation policy. How is it possible to form a table from such rich and continuous observations and hoping to see them again? The observation space should become finite in a way to make it make sense.

----------------------------------------------------
Grammar, typos, etc.:

Line 5: "augments an Gaussian process" -> "augments a Gaussian process"

Line 16: "success in automatically policy learning" -> "success in automatic policy learning"

Line 28: "existing methods can not shed" -> "existing methods cannot shed"

Line 204: "likelihood is computational expensive" -> "likelihood is computationally expensive"

Line 344: "patched policy introduce too many" -> "patched policy introduces too many"

Line 382: "demonstrates the our method’s" -> "demonstrates our method’s"

----------------------------------------------------
References:

[1] Koul, Anurag, Sam Greydanus, and Alan Fern. "Learning finite state representations of recurrent policy networks." arXiv preprint arXiv:1811.12530 (2018).

[2] Danesh, Mohamad H., et al. "Re-understanding Finite-State Representations of Recurrent Policy Networks." International Conference on Machine Learning. PMLR, 2021.

[3] Lin, Yen-Chen, et al. "Tactics of adversarial attack on deep reinforcement learning agents." arXiv preprint arXiv:1703.06748 (2017).

**Time Spent Reviewing:**

3-4 hours

---

> ### Author Response · Authors · 2021-08-10
> **Response to Reviewer mSsX (Part 2).**
>
> **6. Compare our attack in Section 5 with an existing attack.**
>
> The attack in [1] manipulates the observations at the selected time step, whereas our attack directly changes the action at the important steps identified by our method. They cannot be directly compared due to the different attack spaces (observation space vs. action space). To enable the comparison, we applied the time step selection method developed in [1] to choose time steps to modify actions and compared this attack with ours. The results in the following link: https://tinyurl.com/kfxu4zrr show that our attack has a stronger exploitability, confirming the advantage of our method in identifying important steps. We will include these experiment results in the next version.
>
> **7. How to match the current state with those in the look-up table for games with continuous observation spaces.**
>
> In our implementation, we compute the $l_1$ norm difference of the current state $s_t$ and the states ($s_1, …, s_n$) in the table. If the state difference of $s_t$  and $s_i$  is lower than a small threshold (1e-4 in our experiment. We tested 1e-3, 1e-4, and 1e-5 and observed similar results.), we treat $s_t$ and $s_i$ as the same state. Since the games of the same agent usually start from relatively similar states and transition following the same policy, it is possible to observe similar states in different episodes. Note that, in our experiment, we never observe the case where $s_t$ corresponds to multiple states in the table. We will include this description in the next version.
>
> **8. Grammar and typos.**
>
> Thanks for pointing them out. We will address them in the next version.
>
> [1] Tactics of adversarial attack on deep reinforcement learning agents, IJCAI 2017.

---

> > ### Comment · Reviewer_mSsX · 2021-08-12
> > **Response to authors**
> >
> > I would like to thank the authors for their thorough response. It clarified a lot of questions and concerns I have had in mind. However, the paper in its initial form requires many corrections and additions, as promised by the authors. I am willing to raise my rating if the revised version of paper addresses those concerns properly.

---

> > > ### Author Response · Authors · 2021-08-12
> > > **Response to Reviewer mSsX.**
> > >
> > > We thank Reviewer mSsX for the quick response. We are very happy that our response was able to clarify the Reviewer's concerns. Given that we are not allowed to submit new revisions, we will definitely include these corrections in the camera-ready version as soon as it is allowed. In addition, if Reviewer mSsX asks for it and it does not violate the submission rule (We will check with the PC chair about this), we are willing to put together a new version with all the corrections and put it somewhere through a link anonymously. The Reviewer could take a look at it and see if we have addressed the comments properly and if there are new comments. Once again, we thank the Reviewer for the insightful comments and the response.

---

> > > > ### Comment · Reviewer_mSsX · 2021-08-27
> > > > **Updating my rating**
> > > >
> > > > Given the authors response, I increase my rating from 5 to 6. I would like to thank the authors for their thorough discussion.

---

> > > > > ### Author Response · Authors · 2021-08-27
> > > > > **Response to Reviewer mSsX**
> > > > >
> > > > > We thank Reviewer mSsX for the reconsideration. Your review was very detailed and helped us find several points that needed further explanation. We believe the paper will be much improved for having had this discussion, and we will make sure this is reflected in the final revision.

---

> > > ### Author Response · Authors · 2021-08-27
> > > **Follow-up with Reviewer mSsX**
> > >
> > > We thank Reviewer mSsX again for their insightful comments and the positive feedback to our response. Since the discussion phase is about to end and the Reviewer has agreed to raise their score, we are writing to kindly ask if the Reviewer has any additional comments/requirements to update their score. We are at their disposal for any further questions or requirements. We sincerely appreciate the reviewer for their time and consideration.

---

> ### Author Response · Authors · 2021-08-10
> **Response to Reviewer mSsX (Part 1).**
>
> We thank Reviewer mSsX for the constructive and insightful suggestions. Please see below for our response.
>
> **1. Attention is not a good choice for DRL explanation.**
>
> The reviewer commented that using attention for DRL explanations is not a good choice. In our submission, we agree with the reviewer’s comment (see lines 110-112). The reviewer provides more discussion regarding why attention should not be a desired choice for DRL explanations. In the submission, we were aware of these arguments but did not include them in the paper due to the space limit. We thank the reviewer for pointing them out and will add them to our new version.
>
> **2. Comparison with two existing methods.**
>
> The reviewer questioned the statement in lines 56-57 (i.e., we stated, “To the best of our knowledge, this is the first work that interprets a DRL agent’s decision from a sequence of its past actions and states”), and pointed out two papers [1,2]. We carefully read these two papers and noted explanations provided by these two papers are different from our explanation that highlights the time steps critical to the final reward. Below, we explain the difference in detail. To eliminate the confusion created by our statement and these two papers, we will change our statement to “Our work is the first work that interprets a DRL agent’s policy by identifying the most critical time steps to the agent’s final reward in each episode.” in the next version.
>
> Regarding [1], the model proposed in this work can be used to train a more interpretable RNN policy. This policy could identify whether the action at each time step depends more on the current observation or on the previous states (note: not pinpointing the special states as our method did). As a result, this explanation is different from the one delivered by our method (i.e., important time steps w.r.t the final result of each episode). This means users still need our method to extract important time steps even using this model as the policy network. In addition, since the method in [1] is not designed to extract important time steps within an input sequence (As mentioned in Sec. 4.3 of [1], the main focus of this paper is not providing comprehensive explanations), we cannot directly use it to fit the trajectories and extract important time steps (i.e., using it as one of our baselines for comparison). As such, [1] is not suitable for our problem. As future work, we will explore how to combine [1] with our method to further enhance the explainability of a DRL policy.
>
> [2] is a follow-up work of [1]. In [2], the authors first proposed a method to reduce the size of the FSM extracted by [1]. Then, they identified a series of the decision points (i.e., the FSM’s states with multiple child branches) from the reduced FSM and pinpointed the observations leading to different subsequent branches at each decision point. The work [2] treats the identified observations as the important steps. The key difference between [2] and our work is that, in [2], the important steps are w.r.t. the subsequent transitions in the FSM rather than the final result of an episode. This means that, by design, the key steps pinpointed by [2] shouldn’t be equivalent to the key steps truly critical to the final reward. Additionally, [2] was released recently -- not until after our NeurIPS submission. In this rebuttal phase, we contacted the authors of [2] and tried to conduct a comparison. We received the authors’ response without their complete code release (The code requires further cleaning before open-sourcing). Without the code, implementing and replicating the work in [2] requires non-trivial effort and domain knowledge in state machines. We thank the reviewer for pointing this out and will leave this comparison as one of our future works.
>
> In addition to the difference above, there are some other differences distinguishing the works [1,2] from ours. First, both [1] and [2] can be applied only to RNN policies, whereas our method is independent of the policy network’s structures. In our experiments, You-Shall-Not-Pass, CartPole, and Pendulum all use MLP policies. In these games, [1, 2] cannot be applied. Second, the explanations provided by [1] and [2] are mainly used for understanding agent’s policies. Going beyond this utility, Section 5 in our paper further shows that our explanations can be used to facilitate adversarial attacks, policy error patching, and defense against adversarial attacks.
>
> Once again, we thank the reviewer for the comment. We will include these two papers and discuss their difference from our work in the next version of our paper.
>
> **3. Questions about the fidelity metric.**
>
> **Q(1):** Not fitting observations at $(t_i, …, t_{i+l})$ into an RNN policy will affect its memory/hidden states at $t_{i+l+1}$ and thus influence its output action at $t_{i+l+1}$.
>
> A: In our implementation, we fit the observation at each time step into the DRL policies and get the corresponding output. At $(t_{i}, …, t_{i+l})$, we force the agent to choose a random action rather than the action given by the target policy network (See our source code edge-src/pong/utils.py lines 160-168 in our supplementary materials for an example). As such, an RNN policy’s memory will not be truncated and thus its output actions will not be influenced. We will include the above descriptions in the next version.
>
> **Q(2):** How to compute $d_{\text{max}}$?
>
> A: $d_{\text{max}}$ is the maximum absolute reward difference of a game. For games with delayed rewards, it is equal to the final reward range (e.g., if a game has a final reward +1000 for winning and -1000 for loss,  $d_{\text{max}}=2000$). For games with instant rewards, we take the maximum absolute reward change across all the collected trajectories as $d_{\text{max}}$. We will add the above descriptions in the next version.
>
> **Q(3):** Make sure the first term in the fidelity metric won’t undermine the second term.
>
> A: Thanks for the suggestion. We considered it in our design. Specifically, we followed [3] and normalized both $p_{l}$ and $p_{d}$ to [0, 1], so the magnitude of $log(p_{l})$ and $log(p_{d})$ won’t undermine each other. We will emphasize this in the next version.
>
> **4. Questions about (1) subjective descriptions in Section 5, (2) explanation for the same game but different policies, and (3) more games for evaluation.**
>
> About the subjective description:  Thanks for pointing this out. In this rebuttal phase, we followed Reviewer 1XkV’s suggestion and conducted a user study. We will include the result of that user study and lower the tone of our subjective description.
>
> About the same game but different policies: We have conducted experiments and shown the results on this in our submission. In Fig. 3 (b) and Fig. 4 of our paper,  we demonstrate, given the game -- You-Shall-Not-Pass, our method derives different explanations for different policies. As such, the claim in lines 297-298 may not be consistent across different policies given that different policies may have different strategies and behaviors. We will include the above discussion in the next version and make the claim in the lines 297-298 more accurate.
>
> About more games: One of the goals of this work is to examine whether identifying time steps critical to the final reward could facilitate performing episode forensics, launching adversarial attacks, developing effective patches for policy flaws. As such, going beyond some existing DRL explanation works (that choose games only from one kind (e.g., [4], [5], [6])), we chose two kinds of games - Atari and MuJoCo. The games we choose are representative and could support our conclusion. As we discussed in lines 372-384, we will experiment our methods on other games in the future. We thank reviewers’ suggestions and will further detail our plan on other games as part of our future research efforts.
>
> **5. Report the mean and variance of the results in Table 1.**
>
> We followed the reviewer’s suggestion and ran two more groups for each experiment in Section 5 and Supplement Section S4. Due to space limits, we summarize the results in the following link: https://tinyurl.com/kfxu4zrr. Table A1, A2, and A3 show the mean and standard deviation of the results of each experiment in Table 1, Table S5, and Table S6. In addition to the mean and standard deviation, we also conducted a paired t-test to further show the statistical significance of our results. The p-values are shown in Table A4, A5, and A6. The new results are aligned with those in the current paper, i.e., our method demonstrates the best performances in most setups across three use cases - launching adversarial attacks, patching policy errors, and robustifying victim policies. We will include these new experimental results and update Section 5 and S4 in the next version.
>
> [1] Learning finite state representations of recurrent policy networks, ICLR 2019.
>
> [2] Re-understanding Finite-State Representations of Recurrent Policy Networks, ICML 2021.
>
> [3] Real Time Image Saliency for Black Box Classifiers, NIPS 2017.
>
> [4] Visualizing and understanding atari agents, ICML 2018.
>
> [5] Towards Interpretable Reinforcement Learning Using Attention Augmented Agent, NeurIPS 2019
>
> [6] Towards Better Interpretability in Deep Q-Networks, AAAI 2019.

---

### Official Review · Reviewer_1XkV · 2021-07-16

**Rating:** 7
**Confidence:** 4

**Summary:**

This paper presents an explainable method for deep reinforcement learning agents. A gaussian process is used to capture the correlations and the effects between episodes, and is then used to predict the final reward. Further, the authors introduce a method (EDGE) that can provide high-level (strategy-level) explanations of the agent behaviour. Evaluation is done in 5 RL game domains, with results showing higher fidelity against the compared baseline methods.

**Ethical Concerns:**

To the best of my knowledge, this work does not have any immediate ethical concerns.


**Limitations And Societal Impact:**

The main technical limitation lies in the evaluation as explained above. The authors have discussed the societal impact.

**Main Review:**

The proposed method is novel, in that it tries to identify the importance of a time step. This is in contrast to most other Rl explainability methods that are focused on feature/state based explanations. This explainable method enables the understanding of the agents’ policy, and can be used to identify policy errors and weaknesses. The work is positioned well in the explainable RL literature, through a well summarized related section. The technical definitions and equations appear to be sound.

The main weekend I see in this paper is in the limited selection of evaluation domains. It is doubtful whether the strength of the proposed model can be demonstrated in these environments (especially in the pong env), as the time step frames can look much similar to each other. This limitation is exacerbated due to the lack of a user-study to evaluate how useful the time-step importance explanations are against other explainable models. In Section 5, authors give their own interpretations of the explanation frames, but this is subjective and can vary from user to user. I encourage authors to add a discussion to address this issue. Other evaluations that focus on policy errors are reported and explained well.

Quality:
The technical definitions and equations are sound. As mentioned above evaluation suffers from some limitations. I believe these issues can be addressed with a careful commentary on the nature of the interpretability of the proposed method from a user perspective (or better yet a user study).

Clarity:

The Paper is well written and is easy to follow. Methods are defined well enough for the reader to easily follow and understand.

Significance:

This paper has a clear contribution that is in contrast to many other explainable RL methods. The use of importance of the time-step is a novel direction. Although there are somewhat similar methods that use policy summarization [1], this method explicitly focuses on the importance of time-steps. The evaluation hinders this paper from highlighting the importance of the contributions. In its current form, it is not clear how these explanations are better than the other types of RL explainability methods (saliency, policy-based etc). The paper can significantly be strengthened by providing evaluations in other domains (can be other Open AI gym envs) that can have more strategic agents. I do believe this approach has promise and would be of great importance to the larger explainable RL community.



[1] Amir, D., & Amir, O. (2018, July). Highlights: Summarizing agent behavior to people. In Proceedings of the 17th International Conference on Autonomous Agents and MultiAgent Systems (pp. 1168-1176).


**Time Spent Reviewing:**

3

---

> ### Author Response · Authors · 2021-08-10
> **Response to Reviewer 1XkV - user study.**
>
> We thank Reviewer 1XkV for the constructive and insightful comments. Please see our response below.
>
> Previous DRL explanation methods derive interpretations of individual actions by identifying the observation's feature importance in regards to the agent's policy network/value function output. Our work highlights the time steps that are critical to an agent's final result in each episode (e.g., win or loss). In previous research, some researchers also conducted user studies to demonstrate the utility of their explanation methods from human perspectives (e.g., [1, 2] uses interpretation to distinguish well-trained (good) and overfitted (bad) agents). Reviewer 1XkV suggests performing a user study to demonstrate the utility of our explanation method. Below, we describe the efforts we made during this response period. We will include our user study in the next version of the paper.
>
> User Study: we obtained IRB approval (STUDY00018467) and conducted a user study to compare our proposed explanation method with a representative explanation method [1] that pinpoints the input features essential to the agent's individual actions via a saliency method. Specifically, we first recruited 30 participants with different backgrounds in DRL and DRL explanations (4 participants have published paper(s) in DRL explanation; 6 participants have read some papers about DRL explanation; 10 participants have a general understanding of DRL explanation; 10 participants have never heard of DRL explanations.). Then, we presented an online survey to these participants. This survey aims to compare our explanation method with [1] from two perspectives. (1) How well can the explanations generated from the two approaches help a user to pinpoint a good policy? (2) How well can the explanations help a user perform episode forensics and thus understand why an agent fails or succeeds? We briefly describe the design of our user study and the study results below.
>
> (1) Identifying a good policy: Given the representative episodes gathered from two agents of the You-Shall-Not-Pass game (one well-trained and the other overfitted to one specific opponent, i.e., an adversarial agent), we first derived explanations for each of these episodes using the aforementioned interpretation methods. Second, we randomly partitioned the 30 participants into two equally-sized groups and presented the episodes to each group. For Group-A, we also presented the explanations that our method generates. For Group-B, we provided the interpretations that the other method [1] generates. Based on the episodes and their corresponding interpretation, we asked each subject to pinpoint the well-trained agent and asked whether the explanations help identify the good policy. We first discovered that 11 out of 15 participants in Group-A correctly identified the well-trained agent, and 63.6% of them found the explanation helpful. Regarding group-B, 10 out of 15 participants identified the good policy, and 50% of them found the explanation helpful. This result shows that our method demonstrates approximately the same utility as the existing explanations in identifying good/bad policies.
>
> (2) Performing forensics: Given a set of representative episodes gathered from one agent, we used the two above explanation methods to derive explanations for each episode. Then, we present to participants the episodes along with the explanations generated by two different methods. We asked the participants which explanation methods are more beneficial in helping the subject understand why the agent fails/succeeds. We discovered that 21 participants (70%=21/30) chose our method. This discovery implies that interpreting by highlighting critical time steps could better facilitate episode forensics than explaining by highlighting critical input to the action at each step.
>
> Below are more details about our user study:
>
> Survey questions for Group-A: https://forms.gle/SfUCRCWhZEag47gj9.
>
> Survey questions for Group-B: https://forms.gle/Kkj4z4wapCTDqXN76.
>
> Questions 1-3 are the same in both surveys. They ask about the participant’s background and whether he/she understands the game rule and two types of explanations (We found all 30 participants correctly answer Questions 2&3).
>
> Questions 4-7 are about identifying good/bad policies. We present four episodes of the well-trained/overfitted agent together with different explanations to the participants (We count a participant as correctly selecting the good policy only if the participant correctly answered all questions.).
>
> Questions 9-10 are about forensic evaluation. They are the same in both surveys, where the participants are presented with the same videos. Each video shows an episode of the same agent together with explanations derived by our method and [1]. The participants are asked to choose which explanation is more helpful. As mentioned above, we found that 21 out of 30 participants chose our explanations in both questions.
>
> [1] Visualizing and understanding atari agents, ICML 2018.
>
> [2] Highlights: Summarizing agent behavior to people. AAMAS 2018

---

> > ### Author Response · Authors · 2021-08-25
> > **Follow-up with Reviewer 1XkV**
> >
> > Thanks the Reviewer 1XkV again for the insightful comments. Since the discussion phase is about to end, we are writing to kindly ask if the reviewer has any additional comments regarding our response. We are at their disposal for any further questions. In addition, if our user study addresses the reviewer's concern, we would like to kindly ask if the reviewer could reconsider their score.

---

> > ### Comment · Reviewer_1XkV · 2021-08-26
> > **Response to authors**
> >
> > I thank the authors for their response. I appreciate the effort that authors put into conducting the user study, and the results seem to imply that the proposed method is useful against the compared benchmark method. Please also include the relevant statistical tests, comparing the 2 models.

---

> > > ### Author Response · Authors · 2021-08-26
> > > **Response to Reviewer 1XkV's new comments.**
> > >
> > > We thank Reviewer 1XkV for the constructive comments. Following the reviewer’s suggestions, we conducted two statistical tests to demonstrate the statistical significance of our user study results.
> > >
> > > Regarding the first experiment (identifying a good policy), as is mentioned in our previous response, 7 ($11 \times 0.636$) out of 15 participants in Group-A correctly identified the good policy with the help of our explanations, and 5 ($10 \times 0.5$) out of 15 participants in Group-B correctly identified the good policy with the help of the explanations given by the baseline approach [1]. To compare the ability of two explanation methods in facilitating good policy identification, we conducted a two population proportion test [2]. Specifically, we first set the null hypothesis $H_0$ as $p_1 = p_2$, where $p_1$ and $p_2$ are the probability of correctly identifying the good policy according to our explanations and the explanations given by the baseline approach [1]. Then, we computed the sample probability for each group -- Group-A: $\hat{p_1} = 7/15$; Group-B: $\hat{p_2} = 5/15$ and the $z$ statistic, i.e., $z = \frac{(\hat{p_1}-\hat{p_2})}{\sqrt{\hat{p_c}(1-\hat{p_c})(\frac{1}{n}+\frac{1}{n})}}$, in which $\hat{p_c} = (7+5)/30$ is the pooled sample proportion and $n=15$ is the number of participants in each group. Plugging in $\hat{p_1}$ and $\hat{p_2}$, we computed $z = 0.745$. Finally, we computed the percentile $r$ of $z$ in the standard normal distribution and obtained the $p$-value as $2(1-r) = 0.456$. Since the $p$-value is not that small (e.g., $\leq 0.05$), we fail to reject $H_{0}$. This result aligns with our observation in the previous response -- our method demonstrates approximately the same utility as the existing explanations in identifying good policies.
> > >
> > > As for the second experiment (Performing forensics), we conducted a binomial test [2] to demonstrate that our explanation method significantly outperforms the existing method [1] in helping policy forensics. In this test, our null hypothesis $H_{0}$ is $p \leq 0.5$, where $p$ is the probability of choosing our explanation method as more helpful for policy forensics. Then, we computed the percentile $r$ of $Y$ in the binomial distribution $B(30, 0.5)$, where $Y=21$ is the number of participants that chose our method as the more helpful one. Finally, we compute the $p$-value as $1-r = 0.008$. This small $p$-value means we should reject $H_0$, indicating that our method is significantly better than the baseline [1] in facilitating policy forensics.
> > >
> > > [1] Greydanus, Samuel, Anurag Koul, Jonathan Dodge, and Alan Fern. Visualizing and understanding atari agents, ICML, 2018.
> > >
> > > [2] Hogg, Robert V., Elliot A. Tanis, and Dale L. Zimmerman. Probability and statistical inference. Upper Saddle River, NJ, USA:: Pearson/Prentice Hall, 2010.

---

> > > > ### Comment · Reviewer_1XkV · 2021-08-27
> > > > **Response to authors**
> > > >
> > > > Thank you authors for providing the statistical tests, which gives a clearer picture of the utility of the model. I will adjust the score accordingly.

---

> > > > > ### Author Response · Authors · 2021-08-27
> > > > > **Response to Reviewer 1XkV**
> > > > >
> > > > > We thank Reviewer 1XkV for their positive feedback. We are pleased that our responses and additional experiments were able to demonstrate our model utility. We will include the user study as well as the statistical tests in the camera-ready version upon the acceptance of this paper.

---

### Decision · Program_Chairs · 2021-09-27

**Decision:**

Accept (Poster)

**Comment:**

The authors propose a new approach to explaining deep reinforcement learning policies. The approach applies to complex or black box policies, and learns Gaussian Process (GP) model with custom function to learn to predict and assess the importance of state-action pairs on game outcomes. The authors illustrate feasibility of their approach in Atari and MuJoCo domains and demonstrate several application areas.

Initial reviews assessed the contribution as strong, especially because of the method's wide applicability and strong experimental evidence. Several concerns were raised as well, including clarity (e.g., the complexity of the method and need for more precise explanations) and concerns about potentially restrictive assumptions. Suggestions for improvements included additional empirical evaluations, including a proposed user study to test the effectiveness of the approach. The discussion between reviewers and authors has been highly productive, with many concerns addressed. The authors went above and beyond by running the suggested user study during the rebuttal period.

As a result of the productive discussion, all reviewers indicated that their concerns have been largely addressed and reached the consensus to recommend acceptance. The AC agrees with this recommendation. Given that substantial new insights were developed during the discussion, the authors are strongly encouraged to carefully incorporate all suggestions as they prepare the camera ready version.